

# Evaluation of OAFlux datasets based on in situ air-sea flux tower observations over the Yongxing Islands in 2016

Fenghua Zhou[1], Rongwang Zhang[1], Rui Shi[1], Ju Chen[1], Yunkai He[1], Lili Zeng[1], Dongxiao Wang[1*], Qiang Xie[2]

[1]State Key Laboratory of Tropical Oceanography, South China Sea Institute of Oceanology, Chinese Academy of Sciences, Guangzhou 510300, China
[2]Institute of Deep-sea Science and Engineering, Chinese Academy of Sciences, Sanya 572000, China; Laboratory for Regional Oceanography and Numerical Modeling, Qingdao National Laboratory for Marine Science and Technology, Qiangdao, 266237, China

*Correspondence to*: Dongxiao Wang (dxwang@scsio.ac.cn)

**Abstract.** The high-quality Yongxing air-sea flux tower (YXASFT), which was specially designed for air-sea boundary layer flux-related observations, was constructed on Yongxing Island in the South China Sea (SCS). Surface bulk variable measurements were collected during a one-year period from 2016/02/01 to 2017/01/31. The sensible heat flux ($SHF$) and latent heat flux ($LHF$) were further derived via the Coupled Ocean-Atmosphere Response Experiment version 3.0 (COARE3.0) using those variables. This study employed the YXASFT in situ observations to evaluate the Woods Hole Oceanographic Institute (WHOI) OAFlux reanalysis data products in the SCS. The study period was divided into the spring, summer_autumn and winter periods to conduct seasonal comparisons for each variable.

First, the reliability of COARE3.0 data in the SCS was validated using direct turbulent heat flux measurements via an eddy covariance flux (ECF) system. The $LHF$ data derived from COARE3.0 are highly consistent with the ECF measurements with a coefficient of determination ($R^2$) of 0.78. Second, to conduct seasonal comparisons, the overall reliabilities of the bulk OAFlux variables diminish in order from $T_a$, $U$, $Q_a$ to $T_s$ based on a combination of $R^2$ values and biases. OAFlux overestimates (underestimates) $U$ ($Q_a$) throughout the year and provides better estimates of both variables in the winter and spring than in the summer_autumn period, which seems to be highly correlated with the monsoon climate in the SCS. The lowest $R^2$ value is observed between the OAFlux-estimated and YXASFT-observed $T_s$, indicating that $T_s$ is the least reliable product and should thus be used with considerable caution. In terms of the heat fluxes, OAFlux considerably overestimates $LHF$ with an ocean heat loss bias of 52 w/m$^2$ (73% of the observed mean) in the spring, and the seasonal OAFlux $LHF$ performance is consistent with $U$ and $Q_a$. The OAFlux-estimated $SHF$ appears to be poorly representative with enormous overestimations in the spring and winter, while its performance is much better during the summer_autumn period. Third, an analysis reveals that the biases in $Q_a$ are the most dominant factor on the $LHF$ biases in the spring and winter and that the biases in both $Q_a$ and $U$ are responsible for controlling the biases in $LHF$ during the summer_autumn period. The biases in $T_s$ are responsible for controlling the $SHF$ biases, and the effects of biases in $T_s$ on the biases in $SHF$ during the spring and winter are much greater than that in the summer_autumn period.



# 1 Introduction

Exchanges of momentum, heat and water vapor fluxes at the air-sea interface constitute a significant component of air-sea interactions, which affect weather processes and climate change at all scales (Zhu et al., 2002; Persson et al., 2002; Frenger et al., 2013). As the surface that lies beneath the atmosphere, the ocean influences the stability of the atmospheric layer and

the evolution of the atmospheric boundary layer through turbulent exchange (Chelton and Xie, 2010). For example, the water vapor supplied by latent heat transport can determine the regional and even global precipitation (de Cosmo et al., 1996). In addition, sensible heat flux (SHF) and latent heat flux (LHF) at the air-sea interface are both important factors that affect changes in the mixing layer and thermocline (Hogg et al., 2009).

Accurate calculations of regional and global air-sea fluxes play a crucial role in driving marine and atmospheric circulation

models, understanding atmosphere-ocean interactions, and evaluating and assessing numerical weather forecast models (Sun et al., 2003). Currently available air-sea flux datasets (including satellite remote sensing inversion data and reanalysis data) are quite uncertain, as they are mainly derived from inaccurate flux modeling algorithms and uncertainties in the measured values of basic observational quantities involved in the calculation of fluxes (Zeng et al., 1998; Josey, 2001; Smith et al., 2001). In turn, these intrinsic uncertainties limit the ability to assess numerical models based on flux datasets (Yu et al.,

2006). Thus, the appropriate evaluation of a flux dataset is necessary prior to use them in specified study sea area.

The South China Sea (SCS) is mainly controlled by various monsoon systems; it is connected with the western Pacific Ocean and the Indian Ocean through marine and atmospheric processes, and thus, the SCS exhibits potential influences on global climate change as well as regional climate regimes (Wang et al., 2006; Shi et al., 2015). Air-sea interactions in the SCS induce many marine meteorological hazards and greatly affect the transfer of heat and water vapor in regions

throughout South China and Southeast Asia (Yang et al., 2015). Acquiring long-term observations of air-sea fluxes in the SCS can therefore help us to better understand the characteristics and evolutionary behavior of air-sea interactions in the SCS, optimize the parameterization schemes in atmospheric models, and improve long-term weather forecasts and extreme hazardous weather alerts.

To achieve the abovementioned scientific goals, a mesoscale observation network in the Xisha sea area in the northern SCS

was initiated in 2008 (Yang et al., 2015) with the primary ambition of researching air-sea interactions. At present, the observation network includes a surface mooring buoy array, a system of shore-based wave-tide gauges, an automatic weather station, a shore-based air-sea boundary flux tower, and a submerged mooring buoy array. A large dataset comprising in situ observational data was obtained to serve as "ground truth" reference data to quantify the uncertainties within regional model flux products for the SCS.

Many in situ observations and model analysis comparisons have been studied in different oceans around the world, including the Arabian Sea (Weller et al., 1998; Swain et al., 2009), the tropical Pacific Ocean (Weller and Anderson, 1996; Wang and McPhaden, 2001), the northeast Atlantic Ocean (Sun et al., 2003; Yu et al., 2004), the Indian Ocean (Goswami, 2003)and the SCS (Zeng et al., 2009; Wang et al., 2013). Among the world's oceans, the SCS is unique in that it is dominated by





seasonally reversing monsoon winds, including those of the northeast and southwest monsoons. Unfortunately, due to limited field observations of flux-related variables, detailed evaluation studies in the SCS are scarce.

In this study, turbulent SHF and LHF variations as well as numerous bulk variables, including the air temperature ($T_a$), sea surface temperature ($T_s$), air humidity ($Q_a$) and wind speed ($U$), from the Woods Hole Oceanographic Institution (WHOI)

Objectively Analyzed Air-Sea Fluxes (OAFlux) project are compared with high-quality tower-based measurements collected from the Yongxing Islands throughout the northern SCS. This investigation spans a full year from 2016/02/01 to 2017/01/31. Seasonal comparisons of the bulk variables and heat fluxes are described in **Sect. 3**. An overview of the instrumentation on the Yongxing air-sea flux tower (YXASFT) in addition to the data and methodology employed in this paper are introduced in **Sect. 2**. Finally, the summary and conclusions are provided in **Sect. 4**.

## 10  2 Instrumentation, data and methods

### 2.1 Yongxing air-sea flux tower (YXASFT)

The 20-m-tall YXASFT **(Fig. 1a),** which was specially designed for the observation of air-sea boundary layer fluxes, is located approximately 100 m off the northeastern coastline of Yongxing Island ($16.84\,°N$, $112.33\,°E$; **Fig. 2b and 2d**). A gradient meteorological system (GMS) and an eddy covariance flux (ECF) system were mounted on the tower **(Fig. 1c)**. A

CR3000 data logger manufactured by Campbell Scientific Company, USA, is used for data sampling, preprocessing, storage, and transmission. The real-time observation data from the YXASFT are open for access at the website http://mabl.scsio.ac.cn:8040 (login: CSL-CER and password: ruhuna). A data sharing agreement must be signed by the user before the data can be downloaded.

The sensor wiring and data acquisition diagram for the YXASFT is shown in **Fig. 2**. The observational variables within the

GMS include $U$, the wind direction ($W_d$), $T_a$, $Q_a$, the air pressure ($P_a$), the net radiation ($R_n$) and $T_s$. Each parameter is sampled once every second, and 1-, 10- and 30-min averages are recorded and transmitted to the data center in real-time. The ECF system can collect high-frequency turbulent data with a 10-Hz sampling frequency. Successive 30-min fully corrected fluxes of the momentum ($T_{au}$), $SHF$, $LHF$, and $CO_2$ ($F_c$) can be calculated using the online program Easy_flux. The sensors in the YXASFT and their respective measurement specifications are listed in **Table 1**. All of the sensors **(Fig. 1e)** have been

checked via pre- and post-installment calibrations by the National Center of Ocean Standards and Metrology.

### 2.2 Data

The data employed in this study originate from two sources. The in situ datasets comprise observations from the sensors on the YXASFT, and the reanalysis datasets are derived from the OAFlux project. **Table 2** shows various information, including the variable height, time period, data interval and data location, regarding the data adopted in this study.



### 2.2.1 *In situ* data

High-frequency turbulent data ($u$, $v$, $w$, $t$, $\rho_v$) were collected by the ECF system installed at a height of 12 m from 2016/02/01 to 2016/03/29. Direct measurements of turbulent data were further used to calculate the fluxes using the eddy covariance (EC) method in a specified time period (30 min or 60 min). Meanwhile, direct measurements of turbulent fluxes using the

ECF system were used only to verify the applicability of version 3.0 of the Coupled Ocean-Atmosphere Response Experiment (COARE3.0) over the SCS.

The selected 30-min averages of the bulk variables ($U$ measured at a height of 10 m and $T_a$, $Q_a$ and $T_s$ measured at a height of 5 m) used for the bulk flux calculations range from 02/01/2016 to 01/31/2017. Note that $T_s$ was measured using an SI-112 infrared radiation thermometer manufactured by Campbell Scientific Company, USA, installed at a height of 5 m, and

therefore, we consider $T_s$ as representative of the sea surface temperature at a depth of 0.05 m. The value of $Q_a$ was derived using **Eq. 1** as described in COARE3.0 using $T_a$, the relative humidity ($R_h$) and the air pressure ($P_a$). Furthermore, this paper also adopts *SHF* and *LHF* averages within 30-min intervals derived via COARE3.0 using the input observed bulk parameters. The measurement heights of $T_a$ and $Q_a$ in the OAFlux dataset are both 2 m, while the measurement heights for these two parameters on the YXASFT are both 5 m. Thus, prior to conducting a comparison, we corrected the corresponding heights of

the in situ data to correspond to the variable heights in the OAFlux dataset using COARE3.0. In addition, downward longwave radiation (*DLR*) data measured using an NR01 net radiometer manufactured by Hukseflux, Netherlands were used in this paper as an indirect variable to infer the cloud cover in the sky.

$$e_s = 6.112e^{\frac{17.502T_a}{T_a+240.97}(1.0007+3.46P_a*10^{-6})}$$

$$Q_a = \frac{621.97e_s}{(P_a-0.378)R_h/100}$$

(1)

### 2.2.2 Reanalysis data

In this paper, the OAFlux reanalysis data were selected for two reasons. First, a previous study showed that the OAFlux dataset is the most preferable among five different products (i.e., ERA-1, NCEPS, JRA55, TropFlux and OAFlux) with regard to *LHF* data over the SCS (Wang et al., 2017). Second, OAFlux represents the most recently updated data product (as of July 2017) accessible for the study period. OAFlux is an ongoing global flux product compiled by WHOI with a spatial resolution of 1°×1°. OAFlux utilizes an integrated analysis method to combine satellite data with modeling and reanalysis

data, and it employs COARE3.0 to calculate heat fluxes (Yu et al., 2008). In this study, the daily mean OAFlux datasets include $U$, $Q_a$, $T_s$, $T_a$, *LHF* and *SHF*, and YXASFT observations during the same time period were used for a comparison.





## 2.3 Methods

### 2.3.1 Bulk algorithm

The bulk algorithm utilized in this study is based on the Monin-Obukhov similarity theory, which is widely considered to be an advanced bulk algorithm (Fairall et al., 1996). The latest bulk flux algorithm in COARE3.0 was used to calculate the heat

fluxes at the air-sea interface in this paper. To date, this method has achieved reliable calculation accuracies in the wind speed range of 1-10 m/s and has made great progress with regard to heat flux calculation accuracies at high wind speed conditions (Fairall et al., 2003). In this method, the calculation equations for the SHF and LHF can be written as follows:

$$SHF = \rho_a C_p C_h U (T_s - T_a) \tag{2}$$

$$LHF = \rho_a L_e C_e U (Q_s - Q_a) \tag{3}$$

where $\rho_a$ represents the air density, $L_e$ represents the latent heat of evaporation, $C_p$ represents the constant-pressure specific heat, $U$ represents the sea surface wind speed (measured at a height of 10 m in this study), $C_e$ and $C_h$ correspond to the turbulence exchange coefficients for the latent heat and sensible heat, respectively, $Q_s$ and $Q_a$ correspond to the air saturation specific humidity at the sea surface and the air specific humidity near the sea surface, respectively, and $T_s$ and $T_a$ correspond to the sea surface skin temperature and the air temperature near the sea surface, respectively. In **Eqs. 2** and **3**, only $U$, $T_s$, $T_a$

and $Q_a$ are independent measurement variables, while the remainder of the variables must be calculated based on the four independent variables.

### 2.3.2 Eddy covariance method

The EC method is one of the most direct ways to measure and calculate turbulent fluxes (Crawford et al., 1993). Reynolds decomposition is utilized to break raw data down into their means and deviations. Furthermore, the values of *SHF* and *LHF*

can be calculated as the covariances between $w$ and scalar values $(t, \rho_v)$ using the following formulas, respectively:

$$SHF = \rho C_p \overline{w't'} \tag{4}$$

$$LHF = \lambda \overline{w' \rho_v'} \tag{5}$$

where $\rho$ is the dry air density, $C_p$ is the specific heat of dry air at a constant pressure (where 1004.67 Jkg$^{-1}$K$^{-1}$ is used in the calculation), and $\lambda$ is the latent heat ratio of water vapor evaporation. The overbar represents the Reynolds–ensemble average,

and the prime symbol denotes the instantaneous deviation from the ensemble average.

### 2.3.3 Data processing

To match the timescale of the OAFlux daily data, we derived the daily means of the YXASFT-observed bulk variables and heat fluxes by averaging all of the 30-min datasets from each day. In addition, we used bilinearly interpolated OAFlux





values (inversely weighted by the distance) from the surrounding four grid points ($111.5\,°E$, $16.5\,°N$; $112.5\,°E$, $16.5\,°N$; $112.5\,°E$, $15.5\,°E$, $111.5\,°E$, $15.5\,°N$) to represent the corresponding OAFlux value at the YXASFT observation site.

The comparison between the YXASFT and OAFlux datasets (described in **Sects. 3** and **4**) was quantitatively analyzed by using the mean bias (*Bias*, defined in **Eq. 6**), root mean squared error (*RMSE*, defined in **Eq. 7**), coefficient of determination ($R^2$) and linear regressions, respectively.

$$Bias = \frac{1}{N}\sum_{i=1}^{N}(x_i - y_i)$$

(6)

$$RMSE = \sqrt{\frac{1}{N}\sum_{i=1}^{N}(x_i - y_i)^2}$$

(7)

where *x* and *y* denote the OAFlux values and YXASFT observations, respectively.

## 3 Results and discussion

### 3.1 Validation of COARE3.0 using direct ECF measurements

The heat fluxes from both YXASFT and OAFlux used for the comparison herein were derived from COARE3.0. However, the COARE algorithm was originally developed for the Tropical Ocean Global Atmosphere-COARE (TOGA-COARE) experiment in tropical oceans (Fairall et al., 1996), while the reliability of COARE3.0 was verified by (Brunke et al., 2003) using 12 ship cruises over tropical and mid-latitude oceans (between $5\,°S$ and $60\,°N$). The adaptability of OAFlux in the SCS must yet be verified due to its unique geographical location (i.e., it is the largest marginal sea in the northwestern Pacific Ocean) and its monsoon climate system. In this study, the EC fluxes directly measured using the IRGASON ECF system manufactured by Campbell Scientific, USA, were used to validate the performance of COARE3.0 in the SCS. The daily LHF time series in COARE3.0 are basically consistent with those in ECF (**Fig. 3a**) with an $R^2$ value of 0.78 (**Fig. 3c**). COARE3.0 underestimates the LHF with a mean bias of 18.55 w/m$^2$ (19.9% of the ECF mean) relative to direct ECF observations. A larger difference in the LHF measurement occurs when relatively larger LHF values are observed (e.g., 2016/02/07 and 2016/02/25), which can be readily observed in **Fig. 3a**. The precipitation on these days is the most likely explanation for the overestimation in the LHF by the ECF system (Mauder et al., 2006). Although the YXASFT possesses a lack of field precipitation observations, we can speculate that precipitation may have occurred on 2016/02/07 based on a 1.8 ℃ drop in the air temperature and an increase of 13% in the relative humidity within the daily mean. In addition, we spot similar trends on 2016/02/25. In contrast, the *SHF* data pair is far from agreement with an $R^2$ value of 0.03 (**Fig. 3d**). The large variation in the SHF observed using the ECF is not detected within the COARE3.0-derived time series (**Fig. 3b**). Direct heat flux measurements with a 60-day interval obtained using the ECF system show that *SHF* (with a mean value of 23.5 w/m$^2$) is significantly smaller than *LHF* (with a mean of 93.3 w/m$^2$). A small *SHF* magnitude may amplify variations in the time series and reduce the $R^2$ values in scatter plots under the same deviation values. In this comparison, we were more concerned



about the magnitude of correlation in the *LHF* data. Thus, COARE3.0 was considered to be receptive and was used as an appropriate bulk flux algorithm over the SCS.

## 3.2 Evaluation of the OAFlux datasets

OAFlux is a flux product based on a composite algorithm that improves the calculation accuracies of flux-related variables

by using a weighting method for target analysis. However, this method could lead to a time-scale mismatch if the data variables have different data sources (Fairall et al., 2010). It is therefore necessary to evaluate the OAFlux dataset to assess its applicability in the SCS.

### 3.2.1 Time series of the YXASFT observations and OAFlux reanalysis data

Time series of the bulk variables and heat fluxes are given in **Figs. 4** and **5**, respectively. As shown in **Fig. 5**, there is an

obvious overestimation in both *SHF* and *LHF* in OAFlux compared with the YXASFT observations, and this overestimation demonstrates an evident seasonal variation. The time series of *LHF* from the YXASFT observations and OAFlux data show essentially consistent variation trends and agree with one another better during the spring (February to March) and winter (December to January) than during the summer and autumn (April to November) (**Fig. 5b**). The *SHF* variation trend appears to be opposite to that of *LHF*, since the deviations during the winter and spring are clearly larger than those during the

summer and autumn (**Fig. 5a**). For the bulk variables in **Fig. 4**, the OAFlux data maintained a higher consistency with the YXASFT observations with regard to the overall variation trend. Furthermore, $U$ and $Q_a$ seemed to match better during the winter and spring periods, while an overestimation (underestimation) in $U$ ($Q_a$) is more evident during the summer and autumn periods (**Fig. 4a** and **4b**). Some abrupt drops (i.e., variations of 3 to 5 days) in the YXASFT $T_s$ observations were obviously not captured by OAFlux (**Fig. 4d**). In the next section, we divide the annual study period into three periods,

namely, spring (2016/02/01-2016/03/31), summer_autumn (2016/04/01-2016/11/31) and winter (2016/12/01-2017/01/31), to conduct a detailed comparison of their seasonal variations.

### 3.2.2 Comparison of the bulk variables

The heat fluxes from both OAFlux and YXASFT were derived using COARE3.0. Thus, we can further analyze the origin of the seasonal deviations in the heat fluxes by conducting seasonal comparisons of the bulk variables. The scatter plots of $U$,

$Q_a$, $T_s$, and $T_a$ constructed using the YXASFT and OAFlux data for the three separate periods are shown in **Fig. 6**, and a quantitative statistical summary for each variable is listed in **Table 3**.

$U$: The spring, summer_autumn, and winter periods in the Yongxing Islands represent the monsoon transition, southwest monsoon and northeast monsoon periods, respectively. Previous studies indicated that the northeast monsoon in the northern SCS is much stronger than the southwest monsoon (Yan et al., 2005). In this study, the observed mean wind speeds during

the three periods were 6.40, 4.97 and 9.40 m/s. It can be seen from **Fig. 6 (first row)** that the $R^2$ values of $U$ between the OAFlux and YXASFT data during the three periods are 0.90, 0.79 and 0.92. OAFlux overestimates the values of $U$ in the



spring, summer_autumn, and winter periods with mean biases of 0.96 (15% of the YXASFT-observed mean value), 1.19 (24%) and 0.67 m/s (7%), respectively.

$Q_a$: The southwest monsoon is often accompanied by a high water vapor and cloudy skies (Chen et al., 2012). Therefore, the $Q_a$ value during the summer_autumn period was the highest throughout the year with an observed mean of 21.08 g/kg. The

$R^2$ values of $Q_a$ between the OAFlux and YXASFT data during the three periods are 0.81, 0.68 and 0.80 (**Fig. 6 (second row)**). In contrast to $U$, OAFlux exhibits an overall underestimation of $Q_a$ in the spring, summer_autumn, and winter periods with dry biases of 0.33 (2%), 0.75 (4%) and 0.11 g/kg (1%), respectively.

$T_a$: The OAFlux $T_a$ values are highly consistent with the YXASFT observations with $R^2$ values of 0.92, 0.84 and 0.89 in the spring, summer_autumn, and winter periods, respectively (**Fig. 6 (fourth row)**). As shown in **Fig. 4c**, both the seasonal

trends and day-to-day variations are effectively captured in the OAFlux data. The OAFlux reanalyzed $T_a$ data have a warmer bias of 0.52 ℃ (2%) in the spring and colder biases of 0.10 (0.3%) and 0.57 ℃ (2%) in the summer_autumn and winter periods, respectively. Consequently, the OAFlux-estimated $T_a$ can be considered as the most reliable variable in this study.

$T_s$: The OAFlux-estimated $T_s$ captures only the seasonal trend, and the estimates exclude some special synoptic signals, such as abrupt drops during cold air temperatures and typhoons or gradual temperature increases induced by the passage of a

warm eddy. The $R^2$ values of $T_s$ between the OAFlux and YXASFT data are relatively small when compared with those of $U$, $Q_a$ and $T_a$, suggesting that the reliability of the OAFlux-analyzed $T_s$ is generally low. In contrast to $U$ and $Q_a$, the OAFlux $T_s$ performance better in the summer_autumn period ($R^2$=0.70) than in the spring ($R^2$=0.47) and winter ($R^2$=0.54) periods, as shown in **Fig. 6 (third row)**.

In summary, the seasonal performances of the OAFlux-estimated $U$ and $Q_a$ seem to be highly correlated with the monsoon

system in the SCS. This manifests a better performance of the OAFlux-estimated $U$ ($Q_a$) during the spring and winter periods characterized by a stronger (drier) northeast monsoon than during the summer_autumn period characterized by a relatively weaker (wetter) southwest monsoon. The significant difference between the $T_s$ estimates may stem largely from the fact that the OAFlux $T_s$ estimates are retrieved using the Advanced Very High Resolution Radiometer (AVHRR), which is easily affected by the presence of clouds. Therefore, the available OAFlux $T_s$ estimates were dramatically reduced during the

abovementioned special synoptic processes. With the onset of the southwest monsoon, the average total cloud cover, low cloud cover and precipitation all increase throughout the SCS (Yan et al., 2003), and the $T_s$ retrieved via the AVHRR should correspondingly exhibit a lower quality. However, this trend is not observed in the results of this paper. We further utilized in situ observations of the outgoing longwave radiation (*OLR*) to infer the sky cloud cover. There is an evidently greater fluctuation in the *OLR* during the winter and spring periods than in the summer_autumn period, indicating that the winter

and spring seasons possess greater probabilities of cloudy days (**Fig. 7**). This interesting phenomenon may be caused by the fact that the intensity of the summer monsoon in 2016 was weaker than those in preceding years; this hypothesis will be further explored hereafter.





### 3.2.3 Comparison of heat fluxes

The scatter plots of the *LHF* and *SHF* estimates obtained from the YXASFT and from OAFlux during the three periods are shown in **Fig. 8**, and a quantitative statistical summary of each variable is also listed in **Table 3**. Note that an upward (downward) heat flux is positive (negative) in this paper, and a positive (negative) value represents the loss (gain) of ocean heat to (from) the atmosphere.

*LHF*: Compared with the YXASFT observations, the OAFlux-estimated *LHF* is overestimated by a mean bias of 50.95 (70%) in the spring, 42.43 (76%) in the summer_autumn and 63.29 w/m$^2$ (74%) in the winter. The $R^2$ values are 0.80 in the spring, 0.66 in the winter and 0.40 in the summer_autumn (**Fig. 8 (first row)**). This is also consistent with the $R^2$ values for $U$ and $Q_a$, which are the two key input factors in the *LHF* calculations.

*SHF*: Large *SHF* variations during the spring and winter are not evident in the YXASFT-derived *SHF* time series (**Fig. 4e**). Compared to *LHF*, the OAFlux-estimated *SHF* has the smallest $R^2$ values for all three individual periods, as shown in **Table 3** for the spring (0.01), summer_autumn (0.31) and winter (0.14). In comparison, the OAFlux-estimated *SHF* is more reliable during the summer_autumn with a mean bias of 1.07 w/m$^2$ than in the spring (16.83 w/m$^2$) or winter (23.56 w/m$^2$).

Overall, we can infer that the OAFlux-estimated *LHF* product is more reliable during the spring and winter periods than during the summer_autumn period, which is consistent with the key input variables $U$ and $Q_a$, and that the product is further affected by the monsoon system in the SCS. Meanwhile, the *SHF* estimates exhibit opposite characteristics relative to those of *LHF*, as the OAFlux *SHF* product is more credible during the summer_autumn than during the spring and winter periods, which is consistent with the seasonal OAFlux $T_s$ performance and is highly correlated with the cloud cover.

### 3.3 Possible effects of bulk variables on the biases in the SHF and LHF

The values of *SHF* and *LHF* were calculated using **Eqs. 2** and **3**. Thus, possible biases in the *LHF* and *SHF* results are mainly associated with the input bulk variables and the parameterization of the turbulent exchange coefficients in the equations. In this paper, the parameterization scheme is not discussed due to limited space. The relationships among the OAFlux *LHF* bias with $U$, $Q_a$ and $T_s$ were studied extensively by a previous study through years of moored buoy data, automatic weather station (AWS) data and cruise data over different regions in the SCS; it was found that the biases in $Q_a$ dominated the *LHF* biases, followed by the biases in $U$ (Wang et al., 2017). To determine whether similar conclusions exist in this study and to quantify the relationships among the heat flux biases and the bulk variable biases, we constructed scatter plots of the biases in *LHF* (**Fig. 9**) and *SHF* (**Fig. 10**) against the biases in $U$, $Q_a$, $T_s$ and $T_a$. All of the biased data were normalized first to understand their relative importance.

*ΔLHF*: The biases in $Q_a$ are the most dominant factor in determining the biases in *LHF* during the spring and winter with relatively high $R^2$ values of 0.38 in the spring and 0.43 in the winter compared with the other biased bulk variables (**Fig. 8 (first and third columns)**). Both of the $Q_a$ and $U$ biases are responsible for controlling the biases in *LHF* during the summer_autumn period with $R^2$ values of 0.36 and 0.32, respectively (**Fig. 8 (second column)**). Both of the biases in $T_s$ and $T_a$ are negligible control factors on the biases in *LHF*, since their $R^2$ values are all relatively small during the three periods



compared with those of $Q_a$ (**Fig. 8 (third and fourth rows)**). These results reveal effects of biased bulk parameters on the biases in *LHF* similar to those reported in previous studies for the SCS (Wang et al., 2013, 2017).

*ΔSHF*: During the observational period, the biases in $T_s$ were the key factor dominating the biases in *SHF*. The effects of $T_s$ biases on the biased *SHF* during the spring ($R^2$=0.79) and winter ($R^2$=0.72) periods were much larger than that during the

summer_autumn period ($R^2$=0.38), which is also consistent with the fact that OAFlux better estimates $T_s$ in the summer_autumn than in the spring or winter (**Fig. 6 third row**). From **Eq. 2**, *SHF* is largely determined by $T_s$-$T_a$, as shown in **Fig. 5**. OAFlux is unable (able) to capture the variations in $T_s$ ($T_a$) during the spring and winter, thereby causing large fluctuations in $T_s$-$T_a$ and further leading to large variabilities in the OAFlux *SHF* time series.

## 4 Summary and conclusions

Successive air-sea heat flux-related observational data were acquired over the course of a year (2016/02/01-2017/01/31) at the YXASFT in the Yongxing Islands. In this paper, we first used direct heat flux measurements from a high-frequency (10 Hz) ECF system to validate the reliability of the COARE3.0 bulk algorithm in the SCS. Then, seasonal comparisons were conducted for the daily mean surface bulk variables and heat fluxes between the WHOI OAFlux products and YXASFT observations. Finally, the effects of biased bulk variables on the biases in the heat fluxes were presented to determine the

possible sources of the biases in *LHF* and *SHF*. The conclusions are summarized as follows.

The magnitude of the mean of the directly measured *SHF* is small compared with that of *LHF* and can even be ignored in air-sea heat flux interactions during the ECF measurement period. Therefore, we were more concerned with the *LHF* estimation differences between COARE3.0 and the ECF system in this validation. The daily mean *LHF* from COARE3.0 was basically consistent with the ECF measurements with a high $R^2$ and an acceptable bias. Furthermore, if possible

precipitation periods were excluded, the consistency between the COARE3.0 and ECF *LHF* data were better. Thus, the COARE3.0 bulk algorithm was considered to be reliable in this study.

Comparisons of the bulk variables revealed that the reliabilities of the OAFlux datasets diminished in order from $T_a$, $U$, $Q_a$ to $T_s$ based on a combination of $R^2$ values and biases. The performances of the OAFlux-estimated $U$ and $Q_a$ seem to be highly correlated with the monsoon system in the SCS; OAFlux provides a better estimation of $U$ ($Q_a$) in the spring and winter

characterized by a stronger (drier) northeast monsoon than in the summer_autumn characterized by a relatively weaker (wetter) southwest monsoon. Similar to a previous study, this study also indicated that $T_s$ is the least reliable OAFlux product (Sun et al., 2003). The $T_s$ signals during special synoptic process were poorly captured by OAFlux due to the presence of clouds, which affect the recorded AVHRR data. The performance of the OAFlux-estimated $T_s$ is better during the summer_autumn than in the winter or spring due to a reduced cloud cover during the summer monsoon period, which could

be attributable to the fact that the summer monsoon in 2016 was weaker than those in preceding years. With respect to a comparison of the heat fluxes, OAFlux considerably overestimates *LHF* with ocean heat loss biases of 50.95 w/m² (70%) in the spring, 42.43 w/m² (76%) in the summer_autumn and 63.29 *w/m²* (74%) in the winter. Consistent with the key input





variables $U$ and $Q_a$, the OAFlux $LHF$ performance is better during the spring and winter than in the summer_autumn, which is further associated with the monsoon climate in the SCS. The seasonal $SHF$ reliability is coincident with that of $T_s$, as the most poorly reliable $T_s$ estimates lead to the most unreliable $SHF$ estimates with enormous overestimations throughout the year. An analysis of the possible sources of biases in the heat fluxes show that biases in $Q_a$ are the most dominant factor in

determining the biases in $LHF$ during the spring and winter. Meanwhile, both of the biases in $Q_a$ and $U$ are responsible for controlling the biases in $LHF$ during the summer_autumn period. Biases in $T_s$ are responsible for controlling the biases in $SHF$, and the effects of biases in $T_s$ on the biases in $SHF$ during the spring and winter are much greater than that in the summer_autumn period.

In summary, estimates of both $T_s$ and $SHF$ using OAFlux should be utilized with considerable caution in further research,

including driving regional ocean models for the SCS. Additionally, $U$, $Q_a$ and $LHF$ should be used with proper consideration due to their seasonal reliability variations. Researchers should feel more at ease using these data during the northeast monsoon than in the southwest monsoon. The performance of the OAFlux-estimated $T_a$ seems to change little with the seasons and is highly consistent with the YXASFT observations throughout the year. Improving the observation capability of the AVHRR sensor under cloudy conditions is necessary for improving the accuracy of $T_s$ estimates and the reliability of

calculating $SHF$. Larger quantities of in situ bulk variable observations and direct turbulent heat flux measurements as well as improvements in the parameterization of variables in different regions of the SCS are also essential for improving the reliability of OAFlux datasets in the SCS.

**Author contribution**

Fenghua Zhou designed the experiments and Rui Shi, Ju Chen and Yunkai He carried them out. Fenghua Zhou and

Rongwang Zhang write the Matlab program and performed the data processing and analysis. Lili Zeng and Dongxiao Wang give help on useful discussions and data collection. Fenghua Zhou and Qiang Xie prepared the manuscript with contributions from all co-authors.

**Competing interests**

The authors declare that they have no conflict of interest

**Acknowledgements**

This study was funded by the National Natural Science Foundation of China (41706102), the Chinese Academy of Sciences (CAS) Key Technology Talent Program of 2016, the Station Network Construction Project-Xisha Marine Observatory of the CAS (KZCX2-YW-Y202), and the Strategic Priority Research Programs of the CAS (XDA11010302 and XDA11010403).



The YXASFT data were provided by the Xisha Deep Sea Marine Environment Observation Station, South China Sea Institute of Oceanology, CAS. All of the in situ data adopted in this study can be obtained by contacting the first author, Fenghua Zhou (zhoufh@scsio.ac.cn). The authors would like to express their gratitude for the reanalysis data products comprising global ocean heat flux and evaporation data provided by the WHOI OAFlux project (http://oaflux.whoi.edu) funded by the NOAA Climate Observations and Monitoring (COM) program. The source code for the COARE 3.0 algorithm is freely available at http://coaps.fsu.edu/COARE/flux_algor/. The first author would like to thank the engineers from Campbell Scientific Company, USA, for their help with the observation system integration and data acquisition on the YXASFT. Finally, the authors thank the anonymous reviewers for their valuable comments and suggestions that improved the quality of this paper.

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





**Figure captions**

**Figure 1. (a) Yongxing Island air-sea flux tower (YXASFT).**

**(b) Google satellite image of Yongxing Island. The red triangle indicates the location of the YXASFT.**

**(c) Instrumentation and data acquisition system mounted on the YXASFT.**

**(d) Map of the northern SCS. The black star indicates the location of Yongxing Island.**

**(e) Pictures of some sensors on the YXASFT.**

**Figure 2. Diagram of the real-time data acquisition system and the sensor wiring scheme on the YXASFT (SEx: single-ended channel; VXx: voltage excitation channel; Px: pulse-input channel; IXx: current excitation channel; SDM: SDM channel; GPRS: General Packet Radio Service; CDMA: code-division multiple access).**

**Figure 3. Daily means of the LHF and SHF time series (top panels) and scatter plots (bottom panels) of COARE3.0 versus ECF (from 2016/02/01 to 2016/03/29). The $R^2$ values, linear regressions and numbers of matched pairs (N) are given in the bottom panels. The solid red line refers to the linear regression of the matched pairs. The solid green line y=x indicates a 1:1 correspondence.**

**Figure 4. Daily mean time series plots of the YXASFT-observed (red solid lines) and OAFlux-analyzed (blue solid lines) U, Qa, Ta,**
**and Ts values over the study period (2016/02/01-2017/01/31).**

**Figure 5. Daily mean time series plots of the YXASFT-observed (red solid lines) and OAFlux-analyzed (blue solid lines) SHF and LHF over the study period (2016/02/01-2017/01/31).**

**Figure 6. Scatter plots of the YXASFT and OAFlux wind speeds at 10 m (U), air specific humidities at 2 m (Qa), and sea surface temperatures (Ts) and air temperatures at 2 m (Ta) during the spring (left column), summer _autumn (middle column), and**
**winter (right column) periods. The units for U, Qa, Ts and Ta are m/s, g/kg, ℃ and ℃, respectively. The linear regression equation, coefficient of determination ($R^2$), and number of matched pairs (N) are given in each panel. The solid red line refers to the linear regression of the matched pairs.**

**Figure 7. Daily mean time series plots of the YXASFT-observed downward long radiation (DLR) over the study period (2016/02/01-2017/01/31).**

**Figure 8. Same as Fig. 5 but for LHF (first row) and SHF (second row).**

**Figure 9. Scatter plots for the biases of U (ΔU), Qa (ΔQa), Ts (ΔTs), and Ta (ΔTa) with respect to the biases of LHF (ΔLHF). All of the data are normalized to the range of -10 to 10 in this paper. The linear regression equation and coefficient of determination ($R^2$) are given in each panel. The solid red line refers to the linear regression of the matched pairs.**

**Figure 10. Same as Fig. 9 but for the biases in SHF.**




**Tables**

**Table 1. List of sensors installed on the YXASFT and their specifications**

| Parameters | Sensor | Scan interval (Hz) | Averaging interval (min) | Installation height (m) |
|---|---|---|---|---|
| Wind speed and direction | Young 05106 | 1 | 1, 10, 30 | 5, 10, 15, 20 |
| Air temperature and humidity | Vaisala HMP155A | 1 | 1, 10, 30 | 5, 10, 15, 20 |
| Four-component radiation | Hukseflux NR01 | 1 | 1, 10, 30 | 8 |
| Sea surface temperature | Campbell SI-112 | 1 | 1, 10, 30 | 5 |
| Eddy turbulent fluxes $(u, v, w, t, \rho_v, T_{au}, SHF, LHF, F_c)$ | Campbell IRGASON | 10 | 30 | 12 |

**Table 2. Information regarding the adopted in situ and reanalysis data**[*]

| Data | Variables | Location | Height (m) | Interval | Period (day) |
|---|---|---|---|---|---|
| In situ bulk variables | $U$ | | 10 | 30 min | |
| | $Q_a$ | | 5 | 30 min | |
| | $T_s$ | | 0.05 | 30 min | |
| | $T_a$ | | 5 | 30 min | 366 |
| | $DLR$ | | 8 | 30 min | |
| In situ bulk heat fluxes | $SHF$ | | 10 | 30 min | |
| | $LHF$ | 112.33° E, 16.84° N | 10 | 30 min | |
| In situ ECF turbulent data | $u$ | | 12 | 0.1 sec | |
| | $v$ | | 12 | 0.1 sec | |
| | $w$ | | 12 | 0.1 sec | |
| | $t$ | | 12 | 0.1 sec | 57 |
| | $\rho_v$ | | 12 | 0.1 sec | |
| | $SHF$ | | 12 | 30 min | |
| | $LHF$ | | 12 | 30 min | |
| OAFlux bulk variables And heat fluxes | $U$ | | 10 | | |
| | $Q_a$ | 111.5° E, 16.5° N | 2 | | |
| | $T_s$ | 112.5° E, 16.5° N | 0.05 | 1 day | 366 |
| | $T_a$ | 112.5° E, 15.5° N | 2 | | |
| | $SHF$ | 111.5° E, 15.5° N | 10 | | |
| | $LHF$ | | 10 | | |

u: wind speed along the sonic x-axis, v: wind speed along the sonic y-axis, w: wind speed along the sonic z-axis, t: sonic temperature, $\rho_v$: water vapor density. The height of the bulk fluxes derived via COARE3.0 for both in situ data and OAFlux are considered at 10 m.




**Table 3. Quantitative statistical summary based on comparisons between daily YXASFT measurements and daily OAFlux products in the spring, summer_ autumn, and winter Periods**

| Season | Variable | OAFlux mean | YXASFT mean | *RMSE* | *Bias* | $R^2$ | Regression $C_1$ | $C_2$ |
|---|---|---|---|---|---|---|---|---|
| Spring | $U$ (m/s) | 7.36 | 6.40 | 1.36 | 0.96 | 0.90 | 0.89 | 1.66 |
| | $Q_a$ (g/kg) | 15.29 | 15.63 | 1.27 | -0.33 | 0.81 | 0.57 | 6.42 |
| | $T_a$ (°C) | 24.10 | 24.62 | 0.68 | 0.52 | 0.92 | 0.90 | 2.06 |
| | $T_s$ (°C) | 25.12 | 24.65 | 1.29 | 0.46 | 0.47 | 0.32 | 17.27 |
| | $SHF$ (w/m$^2$) | 15.46 | -1.37 | 25.64 | 16.83 | 0.01 | -0.45 | 14.84 |
| | $LHF$ (w/m$^2$) | 123.87 | 72.92 | 63.23 | 50.95 | 0.80 | 1.42 | 20.39 |
| Summer_Autumn | $U$ (m/s) | 6.16 | 4.97 | 1.67 | 1.19 | 0.79 | 0.85 | 1.93 |
| | $Q_a$ (g/kg) | 20.33 | 21.08 | 1.09 | -0.75 | 0.68 | 0.66 | 6.47 |
| | $T_a$ (°C) | 28.86 | 28.95 | 0.43 | -0.10 | 0.84 | 1.00 | -0.09 |
| | $T_s$ (°C) | 29.04 | 29.11 | 0.61 | -0.07 | 0.70 | 0.70 | 8.62 |
| | $SHF$ (w/m$^2$) | 1.65 | 0.51 | 6.33 | 1.07 | 0.31 | 1.10 | 1.02 |
| | $LHF$ (w/m$^2$) | 97.97 | 55.98 | 50.49 | 42.43 | 0.40 | 0.94 | 46.04 |
| Winter | $U$ (m/s) | 10.07 | 9.40 | 0.93 | 0.67 | 0.92 | 0.95 | 1.14 |
| | $Q_a$ (g/kg) | 16.35 | 16.47 | 0.67 | -0.11 | 0.80 | 0.71 | 4.60 |
| | $T_a$ (°C) | 24.91 | 25.48 | 0.67 | -0.57 | 0.89 | 0.90 | 1.95 |
| | $T_s$ (°C) | 25.72 | 25.67 | 0.68 | 0.05 | 0.54 | 0.50 | 12.90 |
| | $SHF$ (w/m$^2$) | 13.83 | 9.73 | 28.85 | 23.56 | 0.14 | -1.59 | -1.62 |
| | $LHF$ (w/m$^2$) | 148.32 | 85.03 | 72.35 | 63.29 | 0.66 | 1.30 | 37.45 |

*OAFlux = $C_1 \times$ YXASFT $+ C_2$



## Figures

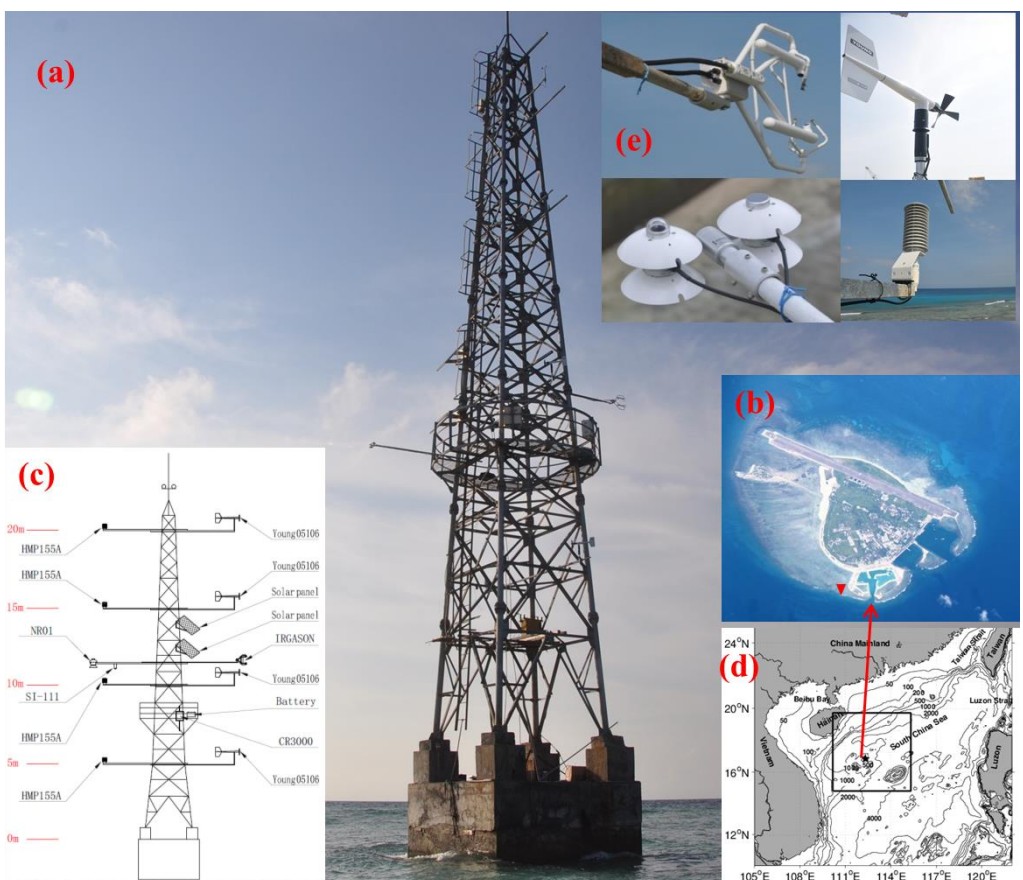

**Figure 1**





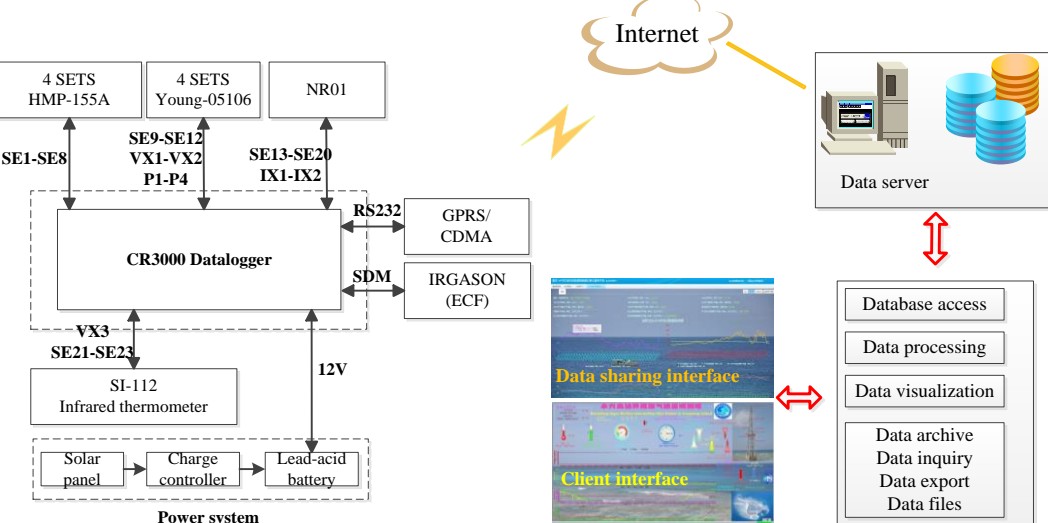

**Figure 2**




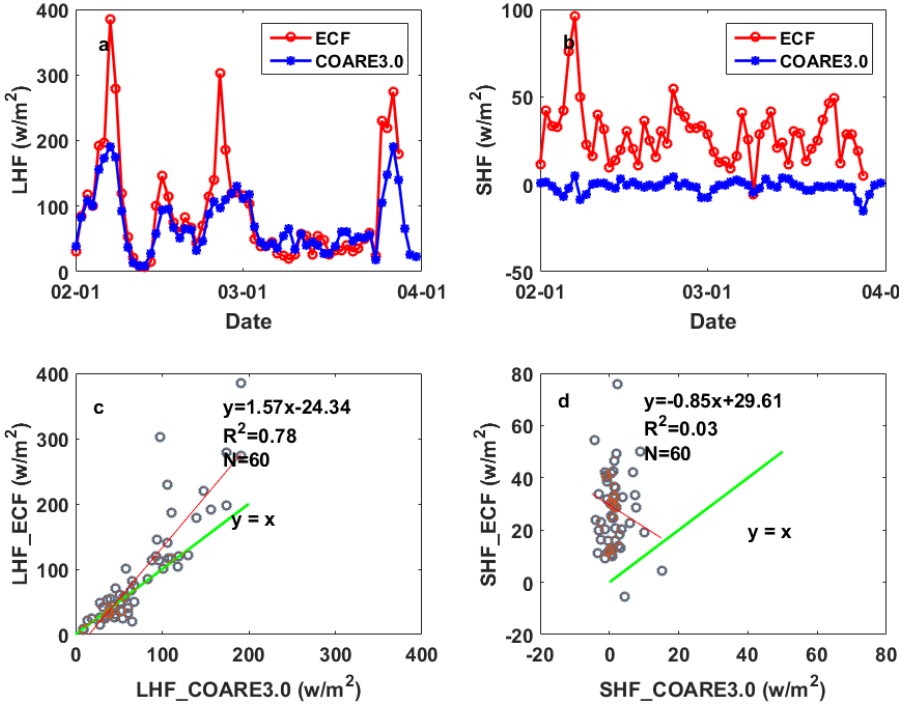

Figure 3



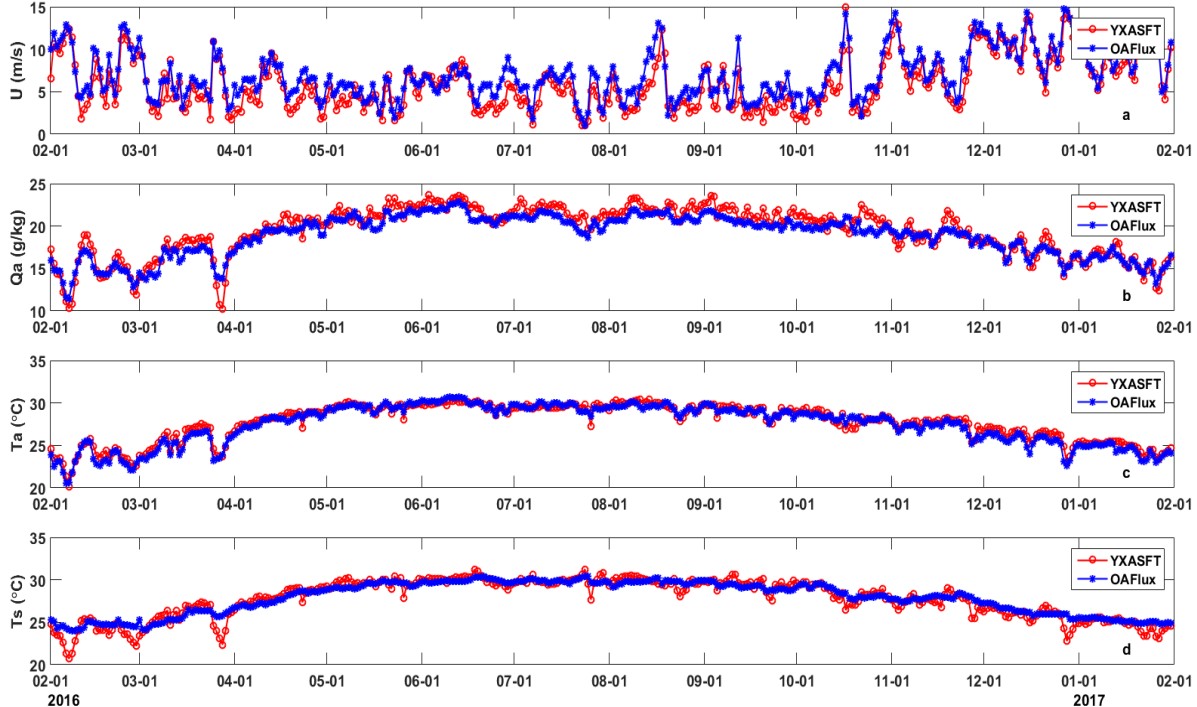

Figure 4

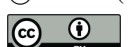



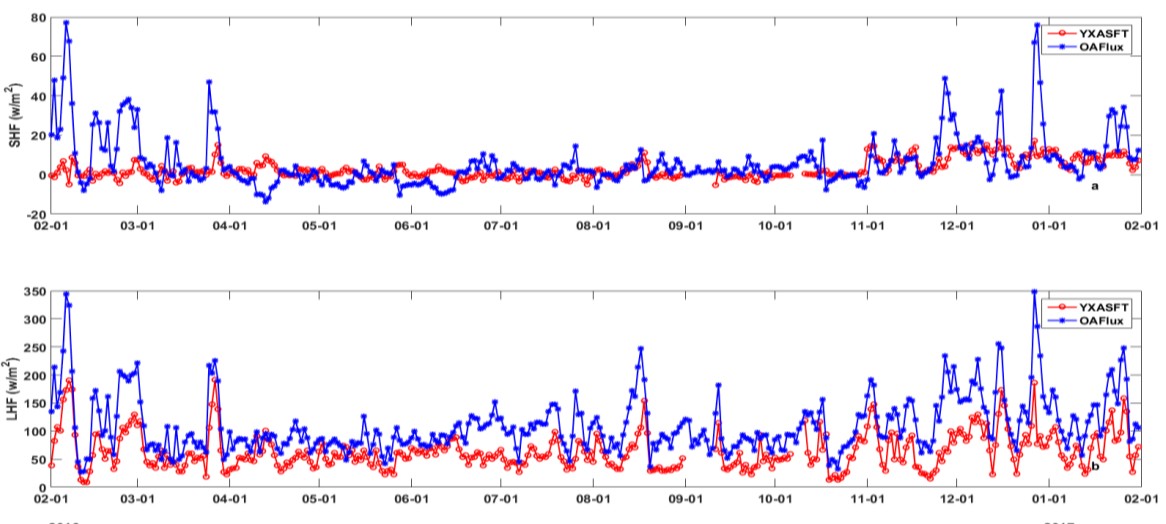

**Figure 5**





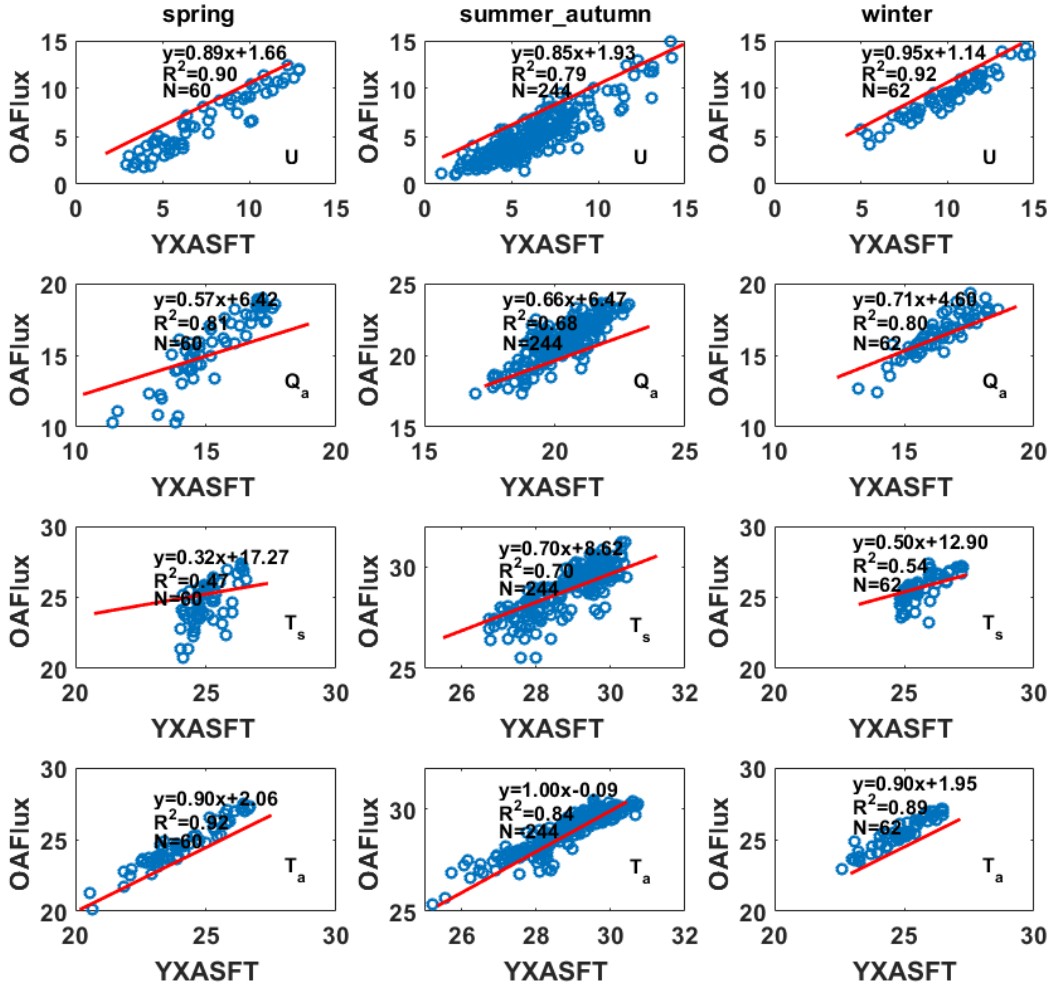

Figure 6

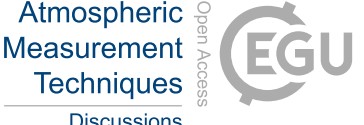



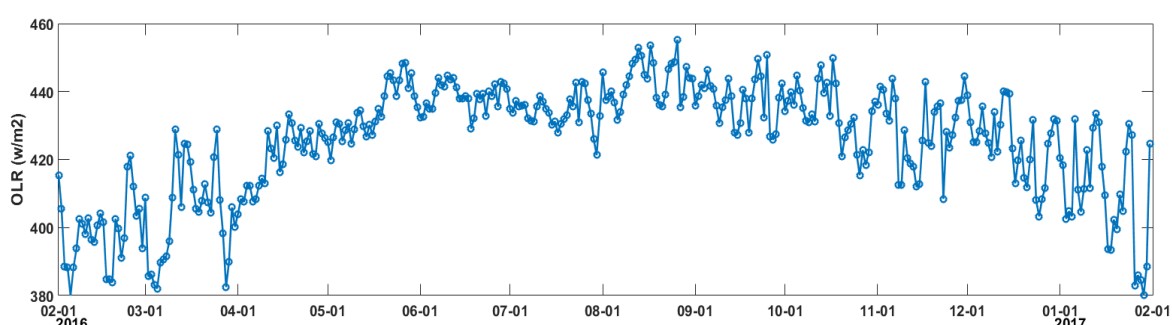

Figure 7

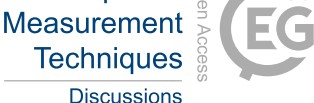



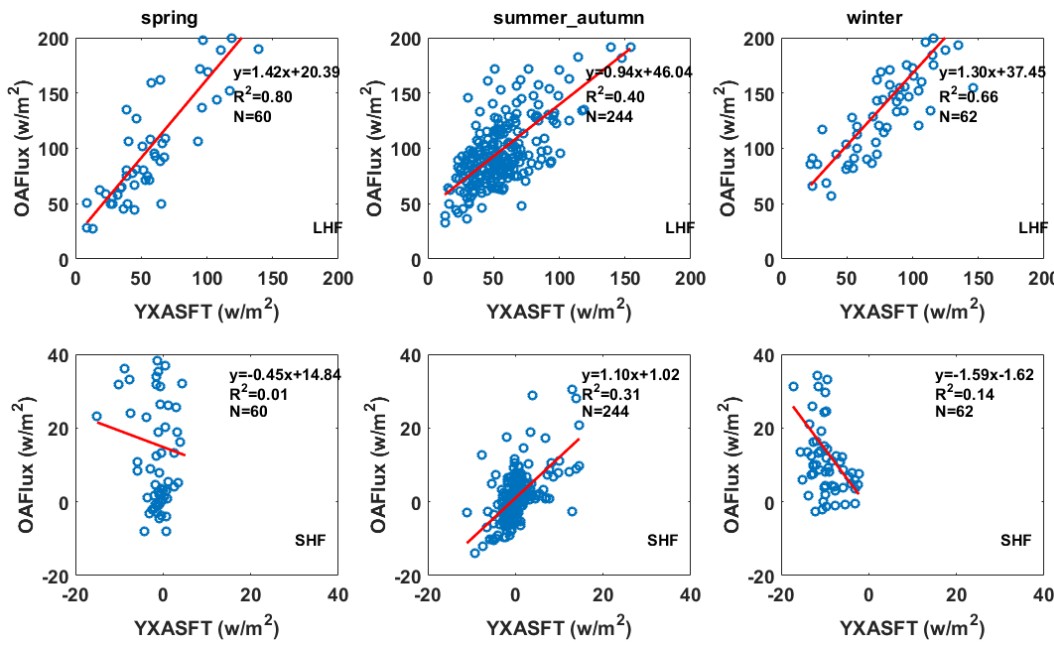

Figure 8



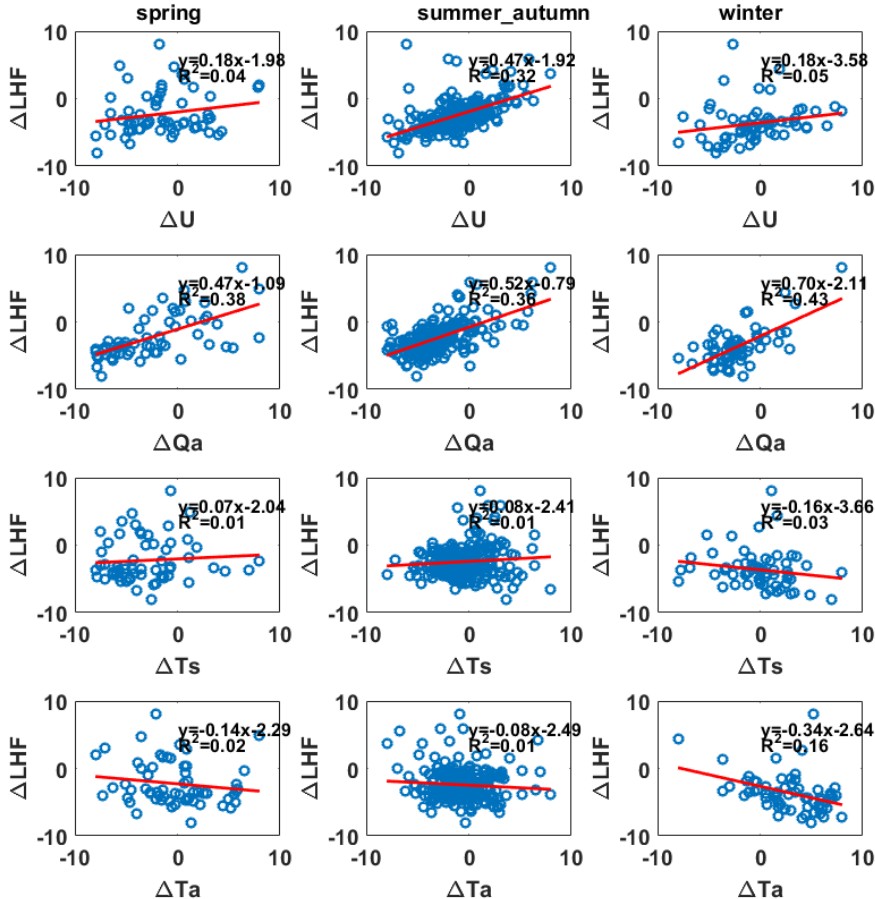

**Figure 9**





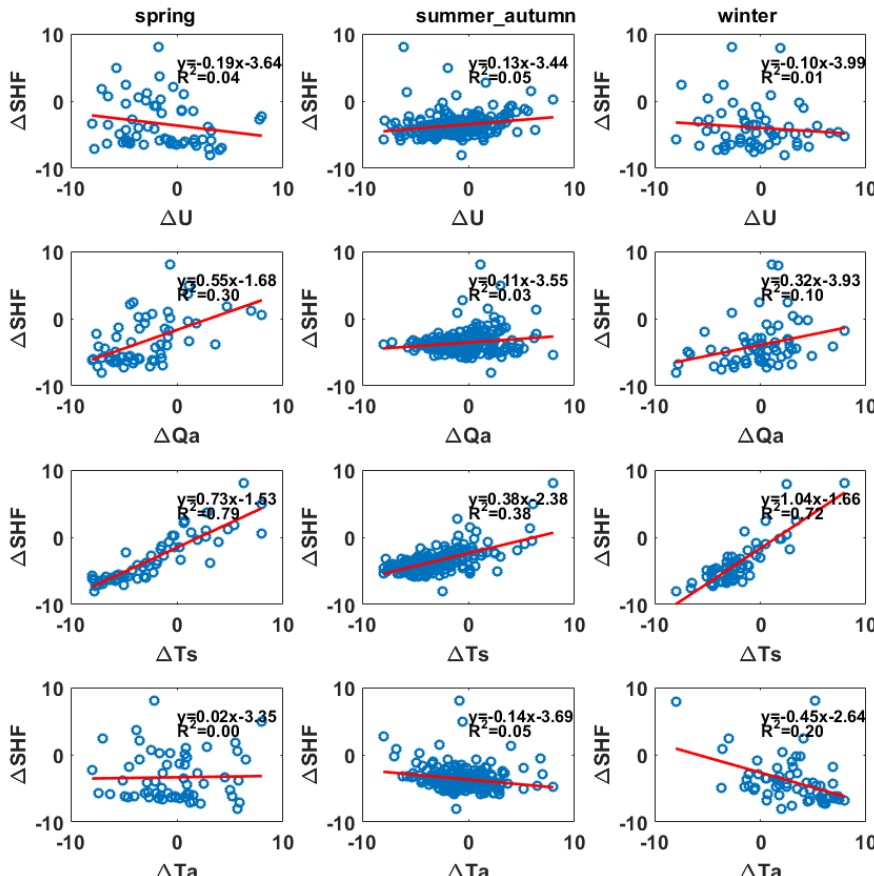

Figure 10