# Peer review of "Evaluation of OAFlux datasets based on in situ air-sea flux tower observations over the Yongxing Islands in 2016"

_Atmospheric Measurement Techniques, 2017_

## Referee Comment (RC1) · Anonymous Referee #1 · 7 Mar 2018

General comments: In this paper, the authors did a nice job to design a high quality air-sea flux tower (YXASFT) in Yongxing Islands for air-sea boundary layer flux-related observations.The instrumentation and the real-time data acquisition system were well designed. Then the authours evaluated the widely used WHOI OAflux reanalysis datasets using in suitdata observedfrom YXASFT. Seasonal comparisons were quantitative analyzed between the OAflux and YXASFT observations by calculating the coefficient of determination, root-mean-square errors, and biases. Through seasonal comparison, the authors get innovative conclusions that the relaibility of OAFlux reanalysis datasets is associated with the monsoon system in SCS, which mainly manifested in the following aspects: 1. OAFlux provides a better estimation of U(Qa) in the spring and winter

characterized by a stronger (drier) northeast monsoon than in the summer_autumn characterized by a relatively weaker(wetter) southwest monsoon 2. The OAFlux LHF performance is better during the spring and winter than in the summer_autumn, whichis further associated with the monsoon climate in the SCS. The authors also quantified the biases in SHF and LHF of the OAFlux datasets and investigated the reasons that may be responsible for the biases. They found that the bias in Qais the main source of error for the LHFin winter monsoon period. Meanwhile, both biased in Qaand U are responsible for controlling the biases in LHF during summer monsoon period. Biases in Ts are responsible for controlling the biases inSHF, and the effects of biases in Ts on the biases inSHF during the spring and winter are much greater than that in thesummer_autumn period.At last, the authors suggest that both Ts and SHF in OAflux are the most unreliable data which should be used with considerable cautions to drive ocean models. Additionally, U,Qa and LHF should be used with proper considerationdue to their seasonal reliability variations. Researchers should feel more at ease using these data during the winter monsoon than in the southwest monsoon. In general, the paper is well-written. Given the importance of the OAflux reanalysis products in the air-sea interaction community, it is worthwhile to systematically evaluate the accuracy of each variable. South China Sea is a region that is lack of sufficient air-sea interaction observations. The authors carried outin suit observations from air-sea tower in this region within relatively long periods, which is of great significance to improve the reliability of reanalysis datasets. The presentation of the results and conclusionsare clear. Thus I recommend the paper to be accepted and published in AMT with minor to moderate revisions. I give the following suggestions to help the authors further improve the paper.

Specific comments: 1. In sec 2.1, the description of sensor wiring and data acquisition system is too simple, with only a single sentence "The sensor wiring and data acquisition diagram for the YXASFT is shown in Fig. 2." in Page 3, line 19. Readers other than professional engineers are difficult to understand this observation system. What's the meaning of SEx, VXx, Px, Ixx. . ., it's a signal, or protocol standard, or sensor hardware interface? I suggest the authors to give more detailed description of the

data acquisition system. For instance, Young-05106 wind sensor with impulse output signal is connected to CR3000 datalogger through Px (channel or protocol standard?). In addition, I pay more attention to the data sharing and data quality. Whether the data can be open access directly by contract the communication author after the publication of the paper? What is the data format? Wheather the necessary data quality control is taken?I visited the data sharing website listed in Page 3, line 17 and found that the web is in Chinese, it's not convinent for non Chinese readers, also I could not found the data download link. 2. In sec 3.1, the authors did a nice job to validate COARE3.0 using the direct eddy convariance flux (ECF) measurements, the verifying results are convincing. However, they didn't give descriptions of the EC data processing steps and the algorithm taken by each step. AsI know that the EC method is mathematically complex, and significant care is required to set up different processing steps for different sites, measurements and study purposes, the difference in the processing algorithm can result in the difference between the turbulent fluxes results. I suggest the authors to add a brief description on how the fluxes are parameterized andcalculatedfor the ECF trubulent data. The authors can also add a figure to express the ECF data processing flow more clearly. For instance, which algorithmswere adopted for coordinatate rotation and WPL compensation? 3. According to the description of in suit data in the paper, I realized that the wind speed range in the YXASFT observed data covers typhoon force winds, as there were at least 2 strong typhoons (No.1603 "MARINAE" and No.1624 "SARIKA")passed through Xisha sea area. So I suggest the authors to add discussions on how COARE3.0 algorithm performs compared to observed exchange coefficients for high wind conditions.

Technical correctionsïjŽ 1. Page 1, line 24, suggest changing"observed" to "calculated". 2. Page 1, line 25,suggest changing"product" to "dataset". 3. Page 1, line 28, delete "an". 4. Page 2, line 7, change "SHF" and "LHF" to "SHF" and "LHF". 5. Page 2, line 12-13, "uncertainties in the measuredvalues of basic observational quantities involved in the calculation of fluxes" this sentence is not clear, suggests change to "uncertainties in the turbulent exchange coefficient were also involved in the fluxes

calculations". 6. Page 2, line 15, change "a flux" to "fluxes". 7. Page 2, line 16, change ";" to ",". 8. Page 2, line 27, change "a shore-based air-sea boundary flux tower" to "a shore-based boundary layer air-sea flux tower". 9. Page 3, line 3, change "SHF" and "LHF" to "SHF" and "LHF". 10. Page 3, line 27, suggest changing"The in situ datasets comprise observations from the YXASFT" to "The in situ observations obtained by YXASFT" 11. Page 4, line 12, isuggest changing"parameters" to "variables". 12. Page 4, line 15, delete"variable". 13. Page 5, line 7, change "SHF" and "LHF" to "SHF" and "LHF". 14. Page 6, line 17,19,20,22,change"LHF" to "LHF". 15. Page 6, line 26, change"SHF" to "SHF". 16. Page 7, line 7, suggest adding"before further application" at the end of the sentence. 17. Page 9, line 19, change "SHF" and "LHF" to "SHF" and "LHF". 18. Page 10, line 5, change "the" to "the". 19. Page 10, line 6, change "or" to " and". 20. Page 11, line 9, "estimates of both Ts and SHF using OAFlux " it not clear, suggest changing to "Both Ts and SHF in OAFlux" 21. Page 11, line 9-10, suggest removing ", including driving regional ocean models for the SCS" as OAFlux is not just used to drive ocean models. 22. Page 21-28, suggest adjusting the colour of the figures to make it unified. 23. Some parts of the text have repeated descriptions. For instance, in Page 2, line 15 "Thus, the appropriate evaluation of a flux dataset in necessary prior to use them in specified study area". In Page 7, line 6-7 "It is therefore necessary to evaluate the OAFlux dataset to assess its applicability in the SCS", these two sentence has almost the same meanning to this paper, no need to repeat in the paper. I suggest the authors to read the article carefully and polish the article to make it more smooth and clear.

Please also note the supplement to this comment:
https://www.atmos-meas-tech-discuss.net/amt-2017-456/amt-2017-456-RC1-supplement.pdf

---

## Short Comment (SC1) · 19 Mar 2018

The high-quality measurements from the YXASFT are extremely precious for the study of SCS-related air-sea exchange processes. The authors have carried out three works: 1) testing the reliability of the COARE3.0 bulk algorithm in the SCS, 2) comparing the seasonal variations between the WHOI OAFlux products and YXASFT observations, and 3) finding the possible sources of the biases in LHF and SHF. These works are very interesting and important for understanding the effectiveness of the presented algorithm (COARE3.0) and data (OAFlux) when using in the SCS, which is helpful for conducting the future SCS-related works. Looking forward to reading the final copy of

this paper.

---

## Author Comment (AC2) · 21 Mar 2018

Thank you for your praise of the article and the recognition of the author's research work. Exactly, what we done in the paper is effectively evaluate the OAflux datasets in the northern of SCS, which will guide the researchers to use OAFlux in a reasonablr way in the SCS-related works. Also we find the possible sources of biases that lead to the biases in the heat fluxes, and give some suggestions for the improvement of future observations to improve the credibility of the OAFlux reanalysis dataset. Anyway, we will continue this study, our next research plan is to improve the parameterization schme in the COARE3.0 bases on the in-suit observations. At last, we will constantly

modify the manuscript until it meets the requirements of AMT, and we will strive for an early publication which will make a contribution to SCS-related air-sea interaction research.

---

## Short Comment (SC2) · 23 Mar 2018

**General Comments:**

The observation data of this work is new and valuable and provides a further validation in the seasonal variability of the accuracy of OAflux products in the South China Sea. It seems that the accuracy of OAflux products varies with the change of prevailing monsoon over the SCS, this is very interesting. The study found that the OAFlux overestimates (underestimates) U (Qa) throughout the year, and the better estimate were found in winter and spring than in summer and autumn. This should be of essential for the air-sea interaction research community in the South China Sea.

**Specific comments:**

The function/aim of COARE3.0 should be given some more explanations. Why you choose COARE3.0 instead of other method to derive SHF and LHF? Authors pointed out that the sea surface temperature Ts is the key variable to determine the differences of sensible heat flux from OAflux products and in-situ observations. It seems that Ts has better accuracy in winter and spring than in summer and autumn. In Page 8, Line 28-30, author mentioned the influence of cloudy days could be the reason for the inaccuracy of Ts derived from AVHRR and the cloud mount can be related to the outgoing long wave radiation (OLR) shown in Fig. 7. OLR is related to the cloud amount, but as I know that OLR is generally obtained by the satellite remote sensing. It cannot be directly observed by the instrument installed on the flux tower introduced in the paper. Besides, the variable shown in Fig. 7 is downward long wave radiation (DLR), so the content of this part is inconsistent and confusion. So, I guess the original intention of the authors is to use the observed DLR to infer the cloud cover. DLR can be used as an indirect variable to infer the cloud cover and its value mainly depends on the air temperature. The larger area cloud cover and the thicker of the clouds will lead to the stronger atmospheric heat preservation and will further result in the strengthening of DLR. Therefore, I suggest the authors to change OLR in Lines 28-30 into DLR to accord with the Fig.7, or use the other OLR datasets derived from satellite observations to infer cloud cover.

**Minor comments:**

The abstract is not concise and coherent enough, and needs to be revised. And, the authors should adjust the range of X-axis, Y-axis and the regression line in Figs 3c and 3d. Same problem appears in Fig 6, Figs 8-10.

The MS need to be edited carefully. There are several obviously misspelled and improperly used words in the MS. For example,

Line 17, 'summer_autumn' should be summer, autumn

Line 21, 'diminish' should be 'were diminished'

Line 21, Definition of 'Ta, U, Qa, Ts' should be given as these variables are first mentioned in the MS.

Line 23, 'summer_autumn period' should be summer-autumn period

Line24, 'is observed' should be 'was observed'

The label of each subset in Figure 1 should be placed in order.

---

## Short Comment (SC3) · 28 Mar 2018

This manuscript evaluates OAFlux datasets using observations collected at an in-situ tower in the SCS in 2016. COARE3.0 algorithm of estimating the SHF and LHF are used to yield the two datasets (OAFlux and YXASFT). Before the comparison between OAFlux and YXASFT, the fluxes of YXASFT from COARE3.0 algorithm were validated by using fluxes from eddy covariance method. Such measurements are rare and valuable. The structure of the manuscript is very good and the content is clear. The results will be valuable for understanding the applicability of OAFlux in the SCS and I recommend publication of the paper. I have only a comment. I suggest authors compare

mean variables (30 min averaged wind, temperature and humidity) obtained from the fast response instruments and slow response instruments. They should be unequal during rain. Because the fast response instruments of Campbell Scientific are very sensitive to precipitation and the observations during precipitation are not available. However, the slow response instruments of YXASFT are available during precipitation. The comparison should be useful for validating the data quality of fast response system.

---

## Referee Comment (RC2) · Anonymous Referee #2 · 4 May 2018

**Review of Manuscript**

**Manuscript ID: amt-2017-456**

Evaluation of OAFlux datasets based on in situ air-sea flux tower observations over the Yongxing Islands in 2016

By

Fenghua Zhou, Rongwang Zhang, Rui Shi, Ju Chen, Yunkai He, Lili Zeng, Dongxiao Wang, and Qiang Xie

**General**

The main aim of this study is to assess the comparisons of latent (LHF) and sensible (SHF) heat fluxes from the high quality Yongxing air-sea flux tower (YXASFT) and OAFlux data. YXASFT LHF and SHF are calculated from bulk variables derived from instrument measurements, while OAFlux fluxes are available as global daily re-analyses with a spatial resolution of 1° in longitude and latitude. The authors handled interesting and needed work aiming at the estimation of heat fluxes. However, the paper requires scientific improvements. I would suggest to further clarify the study objective and the main new findings. The main results, shown in this paper, deal with straightforward comparisons of YXASFT and OAFlux daily flux estimates, with few insights in the physics and the spatial and temporal scale impacts on the comparison results. The paper does not investigate the quality of YXASFT heat fluxes. The results showing the comparison between YXASFT and ECF fluxes are not convincing. The comparisons between the two sources are quite poor. OAFlux flux estimates have been investigated in several papers, including in papers published by the authors. For instance, the bias characterizing mean difference between moorings and OAFlux LHF are quite small. In this study, the LHF biases exhibit "outstanding" values. It would of great interest for scientific community to understand the source of differences between the previous published results and those shown in this manuscript. I am feeling very sorry. I cannot recommend the publication of this paper. However, I strongly encourage the authors to consider the comments aforementioned and listed hereafter for a new enhanced version.

**Specific comments**:

1. Page 3, Line 23 : The correction procedure used for the estimation of Tau, SHF, and LHF should be explained.
2. Page 4: Are bulk variables measured at 20m, 12m, or 10m? The manuscript shows all these values, but does not mention any height correction.
3. Page 3, Line 13: OAFlux are not measurements. They are estimates.
4. Page 5, Lines 18 – 25: It is not clear. Are these calculates handled by the authors or by dedicated online software. The authors mention above the use of Easy-flux software.
5. Page 6, Lines 19 – 24: Do the authors assume that ECF LHF observations are overestimated for rain events?. Does it result from instrumental and/or measurement issues?
6. Page 7, Lines 25 – 28: Convincing scientific and/or technical reasons should be provided for explaining the difference between observed and estimated SHF.
7. Page 7, Lines 11-13: How the YXASFT and OAFlux consistency has been determined?.
8. Page 8, Lines 1-2: The OAFlux U biases are quite high compared to those obtained from moored buoys and OAFlux U10 comparisons. Does this result relies on YXASFT location and/or on OAFlux spatial and temporal resolutions?.
9. Page 8, Lines 22-24: The cloud impact on OAFlux Ts (from NOAA OI SST) should be found everywhere, and especially along tropical are. The previous published studies aiming at the assessment of OAFlux daily data , did not provide Ts results shown in this study.
10. Page 10: The section on top only confirms the results published in several papers. It does provide any new findings dealing with the assessment of LHF and SHF quality or accuracy. Figure 9 and 10 show some interesting results. For instance, the relationship between ΔLHF and ΔTa for winter, would be investigated. Furthermore, the figures show significant scatter. The latter would be investigated as study cases.

---

## Author Comment (AC5) · 25 May 2018

Response to reviewer #2 General comments: The main aim of this study is to assess the comparisons of latent (LHF) and sensible (SHF) heat fluxes from the high quality Yongxing air-sea flux tower (YXASFT) and OAFlux data. YXASFT LHF and SHF are calculated from bulk variables derived from instrument measurements, while OAFlux fluxes are available as global daily re-analyses with a spatial resolution of 1° in longitude and latitude. The authors handled interesting and needed work aiming at the estimation of heat fluxes. However, the paper requires scientific improvements. I would suggest to further clarify the study objective and the main new findings. The

main results, shown in this paper, deal with straightforward comparisons of YXASFT and OAFlux daily flux estimates, with few insights in the physics and the spatial and temporal scale impacts on the comparison results. The paper does not investigate the quality of YXASFT heat fluxes. The results showing the comparison between YXASFT and ECF fluxes are not convincing. The comparisons between the two sources are quite poor. OAFlux flux estimates have been investigated in several papers, including in papers published by the authors. For instance, the bias characterizing mean difference between moorings and OAFlux LHF are quite small. In this study, the LHF biases exhibit "outstanding" values. It would of great interest for scientific community to understand the source of differences between the previous published results and those shown in this manuscript. I am feeling very sorry. I cannot recommend the publication of this paper. However, I strongly encourage the authors to consider the comments aforementioned and listed hereafter for a new enhanced version.

Response: Thank you very much for your review and objective comments on this study. First of all, we should acknowledge that this study focuses more on the in situ measurement techniques and less on the relevant physical processes or other scientific questions. Actually, for a long time in the past, we have been devoted to the construction of YXASFT by installing a varity of observational sensors and uninterrupted maintenance work, with the aim of making it a unique, fixed, multi-parametric and long-term observational tower of air-sea interaction that is still running normally in the open water of the SCS. This study focuses on the introduction of the YXASFT and presenting some preliminary results. To prove the reliability of these in situ observations, we compared the two observational results of high frequency (ECF turbulent flux) and low frequency (bulk flux) at the beginning. In general, the results of LHF in the two sources are in good agreement. Note that some obvious mismatches can be found, which is mainly due to the effect of precipitation of ECF flux data. However, for SHF, variations in the two sources are quite different and big discrepancies exist in them. This partly due to the deficiency of the ECF sensors in the measurement of air temperature, and on the other hand it is related to the defect of the bulk formula in the SHF calculation. We have

explained this with more detail under the response to Specific comments No.6. As one of the most representative flux products, the OAFlux datasets was chosen and compared with full year observations at YXASFT. The YXASFT observations and OAFlux estimates coincide relatively well. On the one hand, this enhanced our confidence on the reliability of YXASFT observations. On the other hand, it helps to find problems in the present flux product and find ways to improve them. Generally speaking, we presented all the problems found in the comparisons and gave possible explanations for every mismatch, which can provide references for YXASFT and OAFlux data users. However, considering the fact that the nature of AMT is focused more on the observation technology, we have not made a deep analysis of the reasons for these problems and related scientific issue. In the future work, with the continuous accumulation of high quality YXASFT observation data, we will focus more on the scientific issues related to the air-sea boundary layer interaction. As the technical director of the YXASFT for its design and maintenance, I have received many requests for data sharing of YXASFT in different private (E-mails, messages from CAS, NUSIT, OUC et al.,) and public occasions (EGU, AGU and AOGS exhibitions). And the publication of this article can greatly enhance our confidence and promote efforts to obtain the in situ observations which are very important to air-sea interaction scientific research around the SCS. At last, we have studied the comments carefully, gave explanation for each questions list below and made some corresponding corrections which we hope meet with your approval.

Specific comments: 1. Page 3, Line 23: The correction procedure used for the estimation of Tau, SHF, and LHF should be explained. Response: Thanks for your suggestion, due to the limited article space, we did not give a detail description of EC data processing step in the paper. The turbulent flux was calculated by an online program named EasyFlux_DL, which was developed by Campbell Scientific Inc, each EC data processing steps we adopted in EasyFlux_DL are as follws: despiking (Vickers et al., 1997), Coordinate rotation (van Dijk et al., 2004), frequency correction (Moncrieff et al., 2004), WPL compensation (Wallace et al., 2006). As suggested by reviewer, we added Figure 3 (Page 22 in the revised paper) to show the EC data correction procedure. And also,

in Page 6, Line 11-14, we have added a description of the EC data processing, as follows: "The EC method is mathematically complex, significant care is required to set up different processing steps for different sites, measurements and study purposes. In this paper, the EC program running on CR3000 was based on the processing steps shown in Fig. 3. "

2: Page 4: Are bulk variables measured at 20m, 12m, or 10m? The manuscript shows all these values, but does not mention any height correction. Response: The U in YXASFT used for comparison were measured in 10m, this is the same height with surface layer U in the OAFlux. The Ta and Qa adopted in YXASFT were measured at 5m, while the measurement height of Ta and Qa in the OAFlux are both 2m. Thus, prior to conducting a comparison, we used the height correction algorithm for bulk variables provided in COARE3.0 to correct the YXASFT observed data to the same height as the bulk variables in OAFlux. This was already explained in the paper (marked by red color), in Page 4, Line 18-20, as follow: "The measurement heights of Ta and Qa in the OAFlux dataset are both 2m, while the measurement heights for these two parameters on the YXASFT are both 5m. Thus, prior to conducting a comparison, we corrected the corresponding heights of the in situ data to correspond to the variable heights in the OAFlux dataset using COARE3.0. "

3. Page 3, Line 13: OAFlux are not measurements. They are estimates. Response: Thank you for reminding us. Yes, OAFlux is an estimated product with the synthesis of reanalysis and satellite inputs. The improper expression and all the similar problems found in the manuscript have been corrected.

4. Page 5, Lines 18 – 25: It is not clear. Are these calculates handled by the authors or by dedicated online software. The authors mention above the use of Easy-flux software. Response: In recent years, many EC data processing methods has been developed and updated, which are mainly divided into two kinds: one is the post-processing software, such as EdiRe, EddyPro, TK3 and so on. The users can use these software to process the direct measured high frequency turbulent data, the built-in correction

algorithm module can be selected with purpose to adapt the location and environment of the observation site. The other is online processing program, such as the Easy_flux developed by Campbell Scientific Inc, which requires the user to preset the adaptive correction algorithm in the program according to the site location and the surrounding environmental condition, the program can calculate the turbulent flux in 30min or 60min in real time. The built in algorithm modules of the online program and post-processing softwares are universally accepted around the world, the calculation results are also very similar. In this paper, we directly use the flux calculation results of the Easy_flux online program to compare with bulk heat fluxes. Further, considering the special location of the island reef and the underlying surface of sea water, we preset a suitable data correction algorithm in order to assure the reliability of the observed data, such as we select planar fitting method for axis correction of sonic wind sensor. A detailed response has been made in Specific comments No.1 in regard to the correction procedure.

5. Page 6, Lines 19 – 24: Do the authors assume that ECF LHF observations are overestimated for rain events? Does it result from instrumental and/or measurement issues? Response: Yes, due to the limitation of the measurement principle of EC sensors, precipitation has great influence on the measurement of high frequency of Qa and Ta (Ta was indirect measured by the ultrasonic, however the principle of ultrasonic measurement of Ta will be seriously affected by precipitation). So, this is a technical problem that has not yet been solved well around the world. Due to there is no direct precipitation observation in the YXASFT, we plot the time series of the 30 min mean variables of U, Ta and Rh in Fig.1 (but this figure was not added in the revised article). As we can see from Fig.1, the time window of four possible precipitations were in 2016/02/03, 2016/02/07, 2016/02/25, 2016/03/26 (marked by dashed ellipse), respectively, which could be obviously shown from a sudden increase in Rh and a sudden drop in Ta. Strictly speaking, the ECF data in these four time windows must be eliminated before compared with COARE3.0. In this paper, we didn't eliminate the possible data polluted by precipitation, but it almost does not affect the validation of LHF. The LHF

comparison between ECF and COARE3.0 shows a good consistency except for the above mentioned possible precipitation windows. We agree very much that if the ECF data during precipitation days were eliminated, the comparison between CAORE3.0 and ECF will be more consistent, which will further demonstrate the reliability of the COARE3.0 algorithm in SCS.

6. Page 7, Lines 25 – 28: Convincing scientific and/or technical reasons should be provided for explaining the difference between observed and estimated SHF. Response: The big difference of SHF between ECF and bulk method can be attributed to both the technical and theoretical reasons. Technical aspects: As we mentioned in the answer of the last question, the Ta was indirectly measured using ultrasonic principle rather than directly physical measurement, so it was easily affected by the precipitation and surrounding environment (Zhang et al., 2016). Further more, Ta was the key factor of SHF calculation and can directly affect the accuracy the SHF in ECF system. So, inaccuracy measurement of Ta by ECF system is a technical problem to be solved. Theoretical aspects: The present bulk method still has large uncertainties in SHF calculation (for example, the uncertainty and limitations of parameterization schemes), which can affect the calculation accuracy of SHF by bulk method. To solve this problem, joint efforts by the scientific community are needed to improve and optimize the parameterization scheme So, on the basis of the technical and theoretical problems mentioned above, the comparison results show that the SHF calculated by ECF and Bulk method is so not consistent with each other. Actually, this problem is well understood as: you can not expect that neither of the two results is accurate enough to have good match with each other. Further, from Fig.9 in the revised paper we can find that the SHF deviation of YXASFT observation and OAFlux is mainly come from spring and winter, but it showed high consistency in the summer_autumn period. This is also consistent with the Ts comparison in Fig.7, which are further affected by the cloud cover in different seasons.

Reference: Zhang R., J. Huang, X. Wang, J. A. Zhang, and F. Huang, 2016: Effects

of precipitation on sonic anemometer measurements of turbulent fluxes in the atmospheric surface layer. J. Ocean Univ. China, 15(3), 389-398.

7. Page 7, Lines 11-13: How the YXASFT and OAFlux consistency has been determined? Response: One side, we try to understand your question from the aspect of spatial matching of the compared datasets. Reply as follows: The YXASFT observation is a signal point, and the OAFlux is a gridded datasets. In order to minimize the uncertainly caused by the location difference, we have adopted the method introduced by (Sun et al, 2003). The representative OAFlux data used for comparison with YX-ASFT is derived by bilinearly interpolated (inversely weighted by distance) from values of the surrounding four grid points. On the other side, we try to understand your question as how to quantify of data consistency from the comparison, and reply as follows: We gave both the time series and the scatter plots of each compared variables in Fig.5 and Fig.7 (in the revised paper), respectively. From Fig.7, the consistency of the two variables can be quantitative analyzed by value of both line coefficient and R2, the bigger value of line coefficient and R2 indicates a better consistency. Reference: Sun B, Yu L, Weller RA (2003). Comparisons of surface meteorology and turbulent heat fluxes over the Atlantic: NWP model analyses versus moored buoy observations. Journal of Climate 16:679–695.

8. Page 8, Lines 1-2: The OAFlux U biases are quite high compared to those obtained from moored buoys and OAFlux U10 comparisons. Does this result relies on YXASFT location and/or on OAFlux spatial and temporal resolutions? Response: Overall speaking, the U10 of OAFlux is in good consistent with the YXASFT observation, with bias of 0.96m/s in Spring, 1.19m/s in Summer_Autumn and 0.67m/s in Winter, and a R2 of 0.90 in Spring, 0.79 in Summer_Autumn and 0.92 in Winter, respectively. However, as shown in Fig.5 and Fig.7 (first row), the U10 in OAFlux is slightly higher than YXASFT observation, and the U10 difference between OAFlux and YXASFT is more obvious in summer. The reason for this may be related to the onset of the summer monsoon and the environmental factors became more complex during this period. The problem

of the larger U10 difference between OAFlux and YXASFT during the monsoon period remains to be further studied. On the other hand, the mismatch in temporal and spatial resolution may also affect the high biases in U. OAFlux is grid data and YXASFT is a single point observation data, the two datasets for comparison can bot be fully spatial matched. So this spatial difference may also lead to the mismatch between of OAFlux and YXASFT observation. The observed daily average data were derived from the average of 48 high-frequency (30 min) observations, but the temporal resolution of the OAFlux's daily average data is not so high (6 hours satellite remote sensing data), so the temporal resolution difference may also lead to the mismatch in their daily average data.

9. Page 8, Lines 22-24: The cloud impact on OAFlux Ts (from NOAA OI SST) should be found everywhere, and especially along tropical are. The previous published studies aiming at the assessment of OAFlux daily data, did not provide Ts results shown in this study. Response: Yes, this suggestion (the available OLR reanalysis data download) has also been given by a Short Comment during the public discussion period. And, we downloaded the daily mean OLR data from NOAA through this web link: https://www.esrl.noaa.gov/psd/cgi-bin/db_search/DBSearch.pl?Dataset=NOAA+Uninterpolated+OLR&Variable=Outgoing+Longwave+Radiation, also we plotted the OLR time series as Fig 2. But from OLR time series, we can not infer that the cloud cover of the sky in winter and spring is more than that during the summer monsoon period (2016/5-2016/9). Then we used DLR (downward longwave radiation) observed from YXASFT to estimate cloud cover indirectly instead of OLR. As we know, DLR is mainly depends on the air temperature, which can be affected by cloud cover. When the sky was covered with large clouds and thick clouds, the probability of rising air temperature will be bigger, which will further increase the DLR. We plotted the curve of observed DLR in Fig.8 (in the revised paper) in the revised paper, from Fig.8 we can see that there is an evidently greater fluctuation in the DLR during the winter and spring periods than in the summer_autumn period, indicating that the winter and spring seasons possess greater probabilities of cloudy

days. Yes, as shown in the previous study, with the onset of the summer monsoon, the sky cloud cover should increase, and the Ts retrieved via the AVHRR should correspondingly exhibit a lower quality. But in this study, we found a different result that the data quality of Ts in OAFlux during the monsoon period is better than that in spring and winter season. And also we have tried to use the observed DLR to explain this phenomenon was caused by the less cloud cover during the summer monsoon period. This interesting phenomenon may be caused by the fact that the intensity of the summer monsoon in 2016 was weaker than those in preceding years, which remains further explored.

10. Page 10: The section on top only confirms the results published in several papers. It does provide any new findings dealing with the assessment of LHF and SHF quality or accuracy. Figure 9 and 10 show some interesting results. For instance, the relationship between ïĄĎLHF and ïĄĎTa for winter, would be investigated. Furthermore, the figures show significant scatter. The latter would be investigated as study cases.

Response: Yes, in terms of biased factor that determine the biased heat flux, we have got some conclusions similar to previous studies. Such as, the biases in Ts were the key factor dominating the biases in SHF and the biases in LHF is mainly caused by the biases in Qa. This does not seem redundant in the article, but proves the credibility of both previous and the present studies. And, we further analyzed the bias factors that dominate the biase of heat flux in different seasons. For example, from Fig.10 (in the revised paper), it can be seen that the LHF bias between YXASFT and OAFlux mainly caused by biased Qa in Spring, biased U and Qa in Summer_Autumn, and bised Qa and Ta in winter, respectively. These dominate factors that cause the seasonal biases in heat flux are new findings in this article. Thank you for your suggestions, we have revised this chapter in Page 9&10, Line 26-33 & 1-3, as follows: $\Delta$LHF: The biases in Qa are the most dominant factor in determining the biases in LHF during the spring with relatively high R2 values of 0.38 compared with the other biased bulk variables (Fig. 10 (first column)). Both of the Qa and U biases are

responsible for controlling the biases in LHF during the summer_autumn period with R2 values of 0.36 and 0.32, respectively (Fig. 10 (second column)). Both of the Qa and Ta biases are the dominate factors in determining the bias in LHF during the winter period with R2 values of 0.43 and 0.16, respectively (Fig. 9 (third column)). The biases in Ts is negligible control factors on the biases in LHF, since their R2 values are all relatively small during the three periods compared with those of Qa (Fig. 9 (third and fourth rows)). In general, the result revealed that the Qa is the most dominated factor controlling the biases in LHF throughout the year is similar to those reported in previous studies (Wang et al., 2013, 2017). Additional, these dominate factors that cause the seasonal biases in LHF are new findings in this article. Yes, it is true that we can find some special phenomena from scatter plots in Fig.10 and Fig.11 (in the revised paper). As you mentioned, from Fig.10, we can see that the relationship of biased LHF and biased Ta in winter is very different from that in spring and summer_autumn, this can be further investigated as a phenomenon study case. This is a good advice, but the main purpose of this paper is to compare the YXASFT observation data and the OAFlux reanalysis data, present the results of comparison objectively, prove the reliability of the observation data, provide references and suggestions for data users. Any in-depth analysis of phenomena or physical process is not described in this paper, but will be further explored in the follow-up research work.

Please also note the supplement to this comment:
https://www.atmos-meas-tech-discuss.net/amt-2017-456/amt-2017-456-AC5-supplement.zip
* * *

---

## Author Response (AR1)

**Response to R1**

**General comments**

In this paper, the authors did a nice job to design a high quality air-sea flux tower (YXASFT) in Yongxing Islands for air-sea boundary layer flux-related observations. The instrumentation and the real-time data acquisition system were well designed. Then the authours evaluated the widely used WHOI OAflux reanalysis datasets using in suit data observed from YXASFT. Seasonal comparisons were quantitative analyzed between the OAflux and YXASFT observations by calculating the coefficient of determination, root-mean-square errors, and biases. Through seasonal comparison, the authors get innovative conclusions that the relaibility of OAFlux reanalysis datasets is associated with the monsoon system in SCS, which mainly manifested in the following aspects: 1. OAFlux provides a better estimation of $U$ ($Q_a$) in the spring and winter characterized by a stronger (drier) northeast monsoon than in the summer_autumn characterized by a relatively weaker (wetter) southwest monsoon 2. The OAFlux $LHF$ performance is better during the spring and winter than in the summer_autumn, which is further associated with the monsoon climate in the SCS.

The authors also quantified the biases in $SHF$ and $LHF$ of the OAFlux datasets and investigated the reasons that may be responsible for the biases. They found that the bias in $Q_a$ is the main source of error for the LHF in winter monsoon period. Meanwhile, both biased in $Q_a$ and U are responsible for controlling the biases in $LHF$ during summer monsoon period. Biases in $T_s$ are responsible for controlling the biases in SHF, and the effects of biases in $T_s$ on the biases in $SHF$ during the spring and winter are much greater than that in the summer_autumn period. At last, the authors suggest that both $T_s$ and $SHF$ in OAflux are the most unreliable data which should be used with considerable cautions to drive ocean models. Additionally, U, $Q_a$ and $LHF$ should be used with proper consideration due to their seasonal reliability variations. Researchers should feel more at ease using these data during the winter monsoon than in the southwest monsoon.

In general, the paper is well-written. Given the importance of the OAflux reanalysis products in the air-sea interaction community, it is worthwhile to systematically evaluate the accuracy of each variable. South China Sea is a region that is lack of sufficient air-sea interaction observations. The authors carried out in suit observations from air-sea tower in this region within relatively long periods, which is of great significance to improve the reliability of reanalysis datasets. The presentation of the results and conclusions are clear. Thus I recommend the paper to be accepted and published in AMT with minor to moderate revisions. I give the following suggestions to help the authors further improve the paper.

**Response:** Thanks for your time, We are pleased to note the favorable comments of anonymous reviewer in his (her) review of our manuscript entitled "Evaluation of OAFlux datasets based on in situ air-sea flux tower observations over the Yongxing Islands in 2016". We have studied theire comments and suggestions carefully and

have made corresponding corrections which we hope meet with his (her) approval. All the corrections are underlined in red and the revised manuscripts was also enclosed as .pdf supplement in AMTD open forum.

**Response to specific comments**

1. As mentioned in the first question of the Specific comments, the reviewer suggest us to add more detailed observation system description mounted on YXASFT.

   Response: The hardware and online data acquisition programme of the observation system was jointly designed by the first author and the CSI (Campbell Scientific, Inc) engineers. We think this part is not very much related to the core study content of the paper, so we just use a brief sentence introduces the system architecture and fig.2 are enclosed for your information. We also applied for the Chinese invention patent of this air-sea boundary layer observation system, if the readers want to know more about this YXASFT and its observation system, can send E-mail to the corresponding author, we will send hardware and data acquasition programme for your reference.

2. What's the meaning of SEx, VXx, Px, Ixx…, it's a signal, or protocol standard, or sensor hardware interface?

   Response: Abbreviations such as SEx, VXx, Px, Ixx appears in Fig.2 are different signal output or input channels in CR3000 datalogger. Take channel SEx for example, it's the channel Signle-Ended Analogy input (CR3000 has 16 channel of SE). I added a reference of CR3000 USER'S MANUAL in Figure 2's caption.

3. In addition, I pay more attention to the data sharing and data quality. Whether the data can be open access directly by contract the communication author after the publication of the paper? What is the data format? Wheather the necessary data quality control is taken?

   Response: Of course, after the publication of the paper, the field observation data used in the YXASFT can be obtained from the authors via email to the corresponding author, we welcome more researchers using the data to verify the conclusions in this paper or carry out more in-depth research study. The data was prvided in the form of .dat, .xls and .txt with a data description header such as:
   Time stamp (YY/MM/DD HH:MM:SS), Wind speed (m/s), Wind direction (°),
   Air temperature (℃) and so on. Yes, we indeed do the necessary data quality control by despiking and averaging. The first hand observation data was obtained directly from data tables stored in CR3000 CF card, CR3000 scan sensors in a 1Hz frequency, an average value (the average value of the 1800 sets of data) was stored in data tables in every 30 mins, despiking was performed before averaging. For each variable, a spike is defined as a value that exceeding window mean $\pm$ 3.5 standard deviations over a certain time window (set to 5 min). Detected spikes

were identified and replaced based on a linear interpolation of neighboring values.

4. I visited the data sharing website listed in Page 3, line 17 and found that the web is in Chinese, it's not convinent for non Chinese readers, also I could not found the data download link.

Response: Thanks for your suggestions and visiting our data website. Yes, the first version of this data sharing website was developed in Chinese language. After all the functions of the website passed running test, we will upgrade this website to both Chinese and English editions. The Login: CSL-CER and password: ruhuna metioned in the paper is only for real-time data and historical data curve display, to download the data files readers should contract the corresponding author for a new authorized username and password and a data sharing agreement must be signed, you can re-login the system with this new username and password, then data download link works.

5. In sec 3.1, the authors did a nice job to validate COARE3.0 using the direct eddy convariance flux (ECF) measurements, the verifying results are convincing. However, they didn't give descriptions of the EC data processing steps and the algorithm taken by each step. As I know that the EC method is mathematically complex, and significant care is required to set up different processing steps for different sites, measurements and study purposes, the difference in the processing algorithm can result in the difference between the turbulent fluxes results. I suggest the authors to add a brief description on how the fluxes are parameterized and calculated for the ECF trubulent data. The authors can also add a figure to express the ECF data processing flow more clearly. For instance, which algorithms were adopted for coordinatate rotation and WPL compensation?

Response: Thanks for your suggestion, in this paper the directly measured eddy convariance heat fluxes by IRGASON ECF system was used to verify the performance of COARE3.0. The online flux calculation program EasyFlux_DL developed by CSI was run in CR3000, the turbulent data processing steps are as follw: despiking [Vickers et al., 1997], Coordinate rotation [van Dijk et al., 2004], frequency correction [Moncrieff et al., 2004], WPL compensation [Wallace et al., 2006].As suggested by reviewer, I added **figure 3** to show the data processing flow of EC data in the paper.

6. According to the description of in suit data in the paper, I realized that the wind speed range in the YXASFT observed data covers typhoon force winds, as there were at least 2 strong typhoons (No.1603 "MARINAE" and No.1624 "SARIKA") passed through Xisha sea area. So I suggest the authors to add discussions on how COARE3.0 algorithm performs compared to observed exchange coefficients for high wind conditions.

Response:
Parameterization of exchange coefficients in high wind is a hot topic in recently years. Both laboratory results [Haus et al., 2010] and field observations [Zhang et al., 2008] indicated that the variations of exchange coefficients in high wind were quite different from that in moderate wind. Some of our observed wind speed reached the high wind level, however, for parametric studies, high frequency observations from several Hz to tens of Hz are essential based on the eddy covariance (EC) method. In this study, the EC system installed on YXASFT only worked for two months from 1st Feb to 29th Mar, and no typhoon passed through Xisha during this period. Consequently, the exchange coefficients could not be obtained from in situ data and the observational LHF was calculated by the bulk method COARE 3.0. Our limited knowledge of the parameterization of exchange coefficients would probably, to some degree, lead to uncertainties in the estimates of LHF under high wind conditions. This is a problem remains to be ameliorated with more available observational data under extreme conditions in the future. It is also an important way to develop more reliable and applicable parameterization schemes for exchange coefficients to improve performances of the LHF products.

Response to technical corrections:
We found the referee's comments on this part are most helpful, we accepted the corrections and suggestion listed from 1st to 21th in the technical corrections, and make corrections in the corresponding place with underlined in red. With regard to the 22th and 23th suggestion, we adjusted the colour of all the figures and deleted the relevant passage since they are not essential to the contents of the paper.

**Response to R2**

**General comments**

The main aim of this study is to assess the comparisons of latent (LHF) and sensible (SHF) heat fluxes from the high quality Yongxing air-sea flux tower (YXASFT) and OAFlux data. YXASFT LHF and SHF are calculated from bulk variables derived from instrument measurements, while OAFlux fluxes are available as global daily re-analyses with a spatial resolution of $1°$ in longitude and latitude. The authors handled interesting and needed work aiming at the estimation of heat fluxes. However, the paper requires scientific improvements. I would suggest to further clarify the study objective and the main new findings. The main results, shown in this paper, deal with straightforward comparisons of YXASFT and OAFlux daily flux estimates, with few insights in the physics and the spatial and temporal scale impacts on the comparison results. The paper does not investigate the quality of YXASFT heat fluxes. The results showing the comparison between YXASFT and ECF fluxes are not convincing. The comparisons between the two sources are quite poor. OAFlux flux estimates have been investigated in several papers, including in papers published by the

authors. For instance, the bias characterizing mean difference between moorings and OAFlux LHF are quite small. In this study, the LHF biases exhibit "outstanding" values. It would of great interest for scientific community to understand the source of differences between the previous published results and those shown in this manuscript. I am feeling very sorry. I cannot recommend the publication of this paper. However, I strongly encourage the authors to consider the comments aforementioned and listed hereafter for a new enhanced version.

**Response:** Thank you very much for your review and objective comments on this study. First of all, we should acknowledge that this study focuses more on the in situ measurement techniques and less on the relevant physical processes or other scientific questions. Actually, for a long time in the past, we have been devoted to the construction of YXASFT by installing variety of observational sensors and uninterrupted maintenance work, with the aim of making it a unique, fixed, multi-parametric and long-term observational tower of air-sea interaction that is still running normally in the open water of the SCS. This study focuses on the introduction of the YXASFT and presenting some preliminary results. To prove the reliability of these in situ observations, we compared the two observational results of high frequency (ECF turbulent flux) and low frequency (bulk flux) at the beginning. In general, the results of LHF in the two sources are in good agreement. Note that some obvious mismatches can be found, which is mainly due to the effect of precipitation of ECF flux data. However, for SHF, variations in the two sources are quite different and big discrepancies exist in them. This partly due to the deficiency of the ECF sensors in the measurement of air temperature, and on the other hand it is related to the defect of the bulk formula in the SHF calculation. We have explained this with more detail under the response to Specific comments No.6. As one of the most representative flux products, the OAFlux datasets was chosen and compared with full year observations at YXASFT. The YXASFT observations and OAFlux estimates coincide relatively well. On the one hand, this enhanced our confidence on the reliability of YXASFT observations. On the other hand, it helps to find problems in the present flux product and find ways to improve them.

Generally speaking, we presented all the problems found in the comparisons and gave possible explanations for every mismatch, which can provide references for YXASFT and OAFlux data users. However, considering the fact that the nature of AMT is focused more on the observation technology, we have not made a deep analysis of the reasons for these problems and related scientific issue. In the future work, with the continuous accumulation of high quality YXASFT observation data, we will focus more on the scientific issues related to the air-sea boundary layer interaction.

As the technical director of the YXASFT for its design and maintenance, I have received many requests for data sharing of YXASFT in different private (E-mails, messages from CAS, NUSIT, OUC et al.,) and public occasions (EGU, AGU and AOGS exhibitions). And the publication of this article can greatly enhance our confidence and promote efforts to obtain the in situ observations which are very important to air-sea interaction scientific research around the SCS.

At last, we have studied the comments carefully, gave explanation for each questions list below and made some corresponding corrections which we hope meet with your approval.

**Specific comments**

1.  Page 3, Line 23: The correction procedure used for the estimation of Tau, SHF, and LHF should be explained.

**Response:** Thanks for your suggestion, due to the limited article space, we did not give a detail description of EC data processing step in the paper. The turbulent flux was calculated by an online program named EasyFlux_DL, which was developed by Campbell Scientific Inc, each EC data processing steps we adopted in EasyFlux_DL are as follws: despiking (Vickers et al., 1997), Coordinate rotation (van Dijk et al., 2004), frequency correction (Moncrieff et al., 2004), WPL compensation (Wallace et al., 2006). As suggested by reviewer, we added **Figure 3** (Page 22 in the revised paper) to show the EC data correction procedure. And also, in Page 6, Line 11-14, we have added a description of the EC data processing, as follows:

"The EC method is mathematically complex, significant care is required to set up different processing steps for different sites, measurements and study purposes. In this paper, the EC program running on CR3000 was based on the processing steps shown in Fig. 3. "

[Figure]

Fig.3 (in the revised paper). EC turbulence data processing and quality control flow chart

2: Page 4: Are bulk variables measured at 20m, 12m, or 10m? The manuscript shows all these values, but does not mention any height correction.

**Response:** The $U$ in YXASFT used for comparison were measured in 10m, this is the same height with surface layer $U$ in the OAFlux. The $T_a$ and $Q_a$ adopted in YXASFT were measured at 5m, while the measurement height of $T_a$ and $Q_a$ in the OAFlux are

both 2m. Thus, prior to conducting a comparison, we used the height correction algorithm for bulk variables provided in COARE3.0 to correct the YXASFT observed data to the same height as the bulk variables in OAFlux. This was already explained in the paper (marked by red color), in Page 4, Line 18-20, as follow:

"The measurement heights of $T_a$ and $Q_a$ in the OAFlux dataset are both 2m, while the measurement heights for these two parameters on the YXASFT are both 5m. Thus, prior to conducting a comparison, we corrected the corresponding heights of the in situ data to correspond to the variable heights in the OAFlux dataset using COARE3.0. "

3. Page 3, Line 13: OAFlux are not measurements. They are estimates.

**Response:** Thank you for reminding us. Yes, OAFLux is an estimated product with the synthesis of reanalysis and satellite inputs. The improper expression and all the similar problems found in the manuscript have been corrected.

4. Page 5, Lines 18 – 25: It is not clear. Are these calculates handled by the authors or by dedicated online software. The authors mention above the use of Easy-flux software.

**Response:** In recent years, many EC data processing methods has been developed and updated, which are mainly divided into two kinds: one is the post-processing software, such as EdiRe, EddyPro, TK3 and so on. The users can use these software to process the direct measured high frequency turbulent data, the built-in correction algorithm module can be selected with purpose to adapt the location and environment of the observation site. The other is online processing program, such as the Easy_flux developed by Campbell Scientific Inc, which requires the user to preset the adaptive correction algorithm in the program according to the site location and the surrounding environmental condition, the program can calculate the turbulent flux in 30min or 60min in real time. The built in algorithm modules of the online program and post-processing softwares are universally accepted around the world, the calculation results are also very similar.

In this paper, we directly use the flux calculation results of the Easy_flux online program to compare with bulk heat fluxes. Further, considering the special location of the island reef and the underlying surface of sea water, we preset a suitable data correction algorithm in order to assure the reliability of the observed data, such as we select planar fitting method for axis correction of sonic wind sensor. A detailed response has been made in Specific comments No.1 in regard to the correction procedure.

5. Page 6, Lines 19 – 24: Do the authors assume that ECF LHF observations are overestimated for rain events? Does it result from instrumental and/or measurement issues?

**Response:** Yes, due to the limitation of the measurement principle of EC sensors, precipitation has great influence on the measurement of high frequency of $Q_a$ and $T_a$ ($T_a$ was indirect measured by the ultrasonic, however the principle of ultrasonic

measurement of $T_a$ will be seriously affected by precipitation). So, this is a technical problem that has not yet been solved well around the world. Due to there is no direct precipitation observation in the YXASFT, we plot the time series of the 30 min mean variables of U, $T_a$ and Rh in Fig.1 (but this figure was not added in the revised article). As we can see from Fig.1, the time window of four possible precipitations were in 2016/02/03, 2016/02/07, 2016/02/25, 2016/03/26 (marked by dashed ellipse), respectively, which could be obviously shown from a sudden increase in Rh and a sudden drop in $T_a$.

Strictly speaking, the ECF data in these four time windows must be eliminated before compared with COARE3.0. In this paper, we didn't eliminate the possible data polluted by precipitation, but it almost does not affect the validation of LHF. The LHF comparison between ECF and COARE3.0 shows a good consistency except for the above mentioned possible precipitation windows. We agree very much that if the ECF data during precipitation days were eliminated, the comparison between CAORE3.0 and ECF will be more consistent, which will further demonstrate the reliability of the COARE3.0 algorithm in SCS.

[Figure]

Fig.1 Time series of observed wind speed (U), air temperature (Ta), air relative humidity (Rh) by the slow and fast response sensors, respectively. The time windows for possible precipitation were marked by dashed ellipse.

6. Page 7, Lines 25 – 28: Convincing scientific and/or technical reasons should be provided for explaining the difference between observed and estimated SHF.

**Response:** The big difference of SHF between ECF and bulk method can be attributed to both the technical and theoretical reasons.

Technical aspects: As we mentioned in the answer of the last question, the $T_a$ was indirectly measured using ultrasonic principle rather than directly physical measurement, so it was easily affected by the precipitation and surrounding environment (**Zhang et al., 2016**). Further more, $T_a$ was the key factor of SHF calculation and can directly affect the accuracy the SHF in ECF system. So,

inaccuracy measurement of $T_a$ by ECF system is a technical problem to be solved.

Theoretical aspects: The present bulk method still has large uncertainties in SHF calculation (for example, the uncertainty and limitations of parameterization schemes), which can affect the calculation accuracy of SHF by bulk method. To solve this problem, joint efforts by the scientific community are needed to improve and optimize the parameterization scheme

So, on the basis of the technical and theoretical problems mentioned above, the comparison results show that the SHF calculated by ECF and Bulk method is so not consistent with each other. Actually, this problem is well understood as: you can not expect that neither of the two results is accurate enough to have good match with each other.

Further, from Fig.9 in the revised paper we can find that the SHF deviation of YXASFT observation and OAFlux is mainly come from spring and winter, but it showed high consistency in the summer_autumn period. This is also consistent with the $T_s$ comparison in Fig.7, which are further affected by the cloud cover in different seasons.

**Response:** Yes, in terms of biased factor that determine the biased heat flux, we have got some conclusions similar to previous studies. Such as, the biases in $T_s$ were the key factor dominating the biases in SHF and the biases in LHF is mainly caused by the biases in $Q_a$. This does not seem redundant in the article, but proves the credibility

of both previous and the present studies. And, we further analyzed the bias factors that dominate the biase of heat flux in different seasons. For example, from Fig.10 (in the revised paper), it can be seen that the LHF bias between YXASFT and OAFlux mainly caused by biased $Q_a$ in Spring, biased U and $Q_a$ in Summer_Autumn, and bised $Q_a$ and $T_a$ in winter, respectively. These dominate factors that cause the seasonal biases in heat flux are new findings in this article. Thank you for your suggestions, we have revised this chapter in Page 9&10, Line 26-33 & 1-3, as follows:

$\triangle$ LHF: The biases in $Q_a$ are the most dominant factor in determining the biases in LHF during the spring with relatively high $R^2$ values of 0.38 compared with the other biased bulk variables (Fig. 10 (first column)). Both of the $Q_a$ and U biases are responsible for controlling the biases in LHF during the summer_autumn period with $R^2$ values of 0.36 and 0.32, respectively (Fig. 10 (second column)). Both of the $Q_a$ and $T_a$ biases are the dominate factors in determining the bias in LHF during the winter period with $R^2$ values of 0.43 and 0.16, respectively (Fig. 9 (third column)). The biases in $T_s$ is negligible control factors on the biases in LHF, since their $R^2$ values are all relatively small during the three periods compared with those of $Q_a$ (Fig. 9 (third and fourth rows)). In general, the result revealed that the $Q_a$ is the most dominated factor controlling the biases in LHF throughout the year is similar to those reported in previous studies (Wang et al., 2013, 2017). Additional, these dominate factors that cause the seasonal biases in LHF are new findings in this article.

Yes, it is true that we can find some special phenomena from scatter plots in Fig.10 and Fig.11 (in the revised paper). As you mentioned, from Fig.10, we can see that the relationship of biased LHF and biased $T_a$ in winter is very different from that in spring and summer_autumn, this can be further investigated as a phenomenon study case. This is a good advice, but the main purpose of this paper is to compare the YXASFT observation data and the OAFlux reanalysis data, present the results of comparison objectively, prove the reliability of the observation data, provide references and suggestions for data users. Any in-depth analysis of phenomena or physical process is not described in this paper, but will be further explored in the follow-up research work.

**Response to SC1**

**Short Comments**

The high-quality measurements from the YXASFT are extremely precious for the study of SCS-related air-sea exchange processes. The authors have carried out three works: 1) testing the reliability of the COARE3.0 bulk algorithm in the SCS, 2) comparing the seasonal variations between the WHOI OAFlux products and YXASFT observations, and 3) finding the possible sources of the biases in LHF and SHF. These works are very interesting and important for understanding the effectiveness of the presented algorithm (COARE3.0) and data (OAFlux) when using in the SCS, which is helpful for conducting the future SCS-related works. Looking forward to reading the final copy of this paper.

**Response:** Thank you for your praise of the article and the recognition of the author's research work. Exactly, what we done in the paper is to effectively evaluate the OAFlux datasets in the northern of SCS, which will guide the researchers to use OAFlux in a reasonable way in the SCS-related works. Also we find the possible sources of biases that lead to the biases in the heat fluxes, and give some suggestions for the improvement of future observations to improve the credibility of the OAFlux reanalysis dataset. Anyway, we will continue this study, our next research plan is to improve the parameterization scheme in the COARE3.0 bases on the in-suit observations. At last, we will constantly modify the manuscript until it meets the requirements of AMT, and we will strive for an early publication which will make a contribution to SCS-related air-sea interaction research.

**Response to SC2**

**General Comments**

The observation data of this work is new and valuable and provides a further validation in the seasonal variability of the accuracy of OAFlux products in the South China Sea. It seems that the accuracy of OAFlux products varies with the change of prevailing monsoon over the SCS, this is very interesting.
The study found that the OAFlux overestimates (underestimates) U (Qa) throughout the year, and the better estimate were found in winter and spring than in summer and autumn. This should be of essential for the air-sea interaction research community in the South China Sea.

**Response:** Thanks for your comments on our MS entitled "Evaluation of OAFlux datasets based on in situ air-sea flux tower observations over the Yongxing Islands in 2016", we have studied your comments carefully and found your comments are very helpful, especially you found out an inconsistent between the Fig.7 and the corresponding description in the paper. We have revised the manuscript according to your comments, and the revised parts were underlined in red. We kindly remind you that the revised manuscript (2nd) is modified based on the revised manuscript (1st) of the first reviewer's opinion.

**Response to Specific comments**
1. The function/aim of COARE3.0 should be given some more explanations. Why you choose COARE3.0 instead of other method to derive SHF and LHF?
**Response**: Due to the limited text, there is no specific description of the COARE3.0 algorithm in this paper. Readers can read the following references for more information of COARE3.0 (Fairall et al., 2003; Lisan et al., 2008). Compared to COARE2.5, the updated COARE3.0 has some noted improvements. The range of wind speed validity is now extended to 0–20 ms$^{-1}$ after modifying roughness representation. The transfer coefficients are redefined in terms of conservative quantity rather than the measured quantity, thus eliminating the need for a Webb et al.

(1980) correction to latent heat flux. The COARE 3.0 is shown to be accurate within 5% for wind speeds of 0–10 m s$^{-1}$ and 10% for wind speeds between 10 and 20 m s$^{-1}$.

There are several forms of bulk flux algorithms currently available (Brunle et al., 2002). The differences between the algorithms reside in the differences in treating the parameterizations of the transfer coefficients $C_e$ and $C_h$, conditions of light wind and stable stratification, influence of sea spray, treatment of sea state (swell, directional effects), appropriate averaging scales, parameterization of mesoscale gustiness, and the behavior of scalar sublayer transfer. In this paper, the OAFlux reanalysis *SHF* and *LHF* data were calculated by the state-of-the-art COARE bulk flux algorithm version 3.0, in order to avoid the deviation caused by different algorithms in the process of comparison and evaluation, so we also adopted COARE3.0 to derive *SHF* and *LHF* to keep consistent with OAFlux.

2. Authors pointed out that the sea surface temperature Ts is the key variable to determine the differences of sensible heat flux from OAFlux products and in-situ observations. It seems that Ts has better accuracy in winter and spring than in summer and autumn. In Page 8, Line 28-30, author mentioned the influence of cloudy days could be the reason for the inaccuracy of Ts derived from AVHRR and the cloud mount can be related to the outgoing long wave radiation (OLR) shown in Fig. 7. OLR is related to the cloud amount, but as I know that OLR is generally obtained by the satellite remote sensing. It cannot be directly observed by the instrument installed on the flux tower introduced in the paper. Besides, the variable shown in Fig. 7 is downward long wave radiation (DLR), so the content of this part is inconsistent and confusion.

**Response:** Thank you very much to point out this inconsistent in the text. Actually, at the begging the authors thought that using OLR to infer the cloud cover directly. However, no remote sensing OLR datasets available during this period of observation was found. We used DLR observed from YXASFT to estimate cloud cover indirectly instead of OLR. As we know, DLR is mainly depends on the air temperature, which can be affected by cloud cover. When the sky was covered with large clouds and thick clouds, the probability of rising air temperature will be bigger, which will further increase the DLR. We plotted the curve of observed DLR in fig.7, but forget to modify the corresponding in the article. Thanks again for reading this paper carefully and find this confusion, we are pleased accept your suggestions, and change OLR into DLR in the article.

**Response to Minor comments:**

**1.** The abstract is not concise and coherent enough, and needs to be revised.
**Response:** We have already revised the abstract based on #1 Reviewer's comments. You could read the new abstract from the revised manuscript in the supplement.
2. The authors should adjust the range of X-axis, Y-axis and the regression line in Figs 3c and 3d. Same problem appears in Fig 6, Figs 8-10.
**Response:** The figures after adjusted were shown on the revised manuscript.
3. Line 21, 'diminish' should be 'were diminished'
Line 21, Definition of 'Ta, U, Qa, Ts' should be given as these variables are first

mentioned in the MS.

Line24, 'is observed' should be 'was observed'

The label of each subset in Figure 1 should be placed in order.

**Response:** The above mentioned suggestions were accepted and revised in the corresponding place of the article

**Response to SC3**

**Short Comments**

This manuscript evaluates OAFlux datasets using observations collected at an in-situ tower in the SCS in 2016. COARE3.0 algorithm of estimating the SHF and LHF are used to yield the two datasets (OAFlux and YXASFT). Before the comparison between OAFlux and YXASFT, the fluxes of YXASFT from COARE3.0 algorithm were validated by using fluxes from eddy covariance method. Such measurements are rare and valuable. The structure of the manuscript is very good and the content is clear. The results will be valuable for understanding the applicability of OAFlux in the SCS and I recommend publication of the paper. I have only a comment. I suggest authors compare mean variables (30 min averaged wind, temperature and humidity) obtained from the fast response instruments and slow response instruments. They should be unequal during rain. Because the fast response instruments of Campbell Scientific are very sensitive to precipitation and the observations during precipitation are not available. However, the slow response instruments of YXASFT are available during precipitation. The comparison should be useful for validating the data quality of fast response system.

**Response:** As you mentioned in the short comments, the fast response eddy flux sensors are very sensitive to precipitation and the observations during the precipitation are not available. Yes, we should pay much attention to this problem in ECF's data quality control, the eddy flux measured by ECF during precipitation must be rejected for further comparison and research. However, due to the limitation of scientific research funds, no precipitation observation equipment is installed on the YXASFT, and the eddy flux (SHF and LHF) measured by ECF system was directly used to compare with COARE3.0.   So, in the chapter 3.1 of this paper, we illustrated this problem with the description of "A larger difference in the LHF measurement occurs when relatively larger LHF values are observed (e.g., 2016/02/07 and 2016/02/25), which can be readily observed in **Fig. 3a**.The precipitation on these days is the most likely explanation for the overestimation in the LHF by the ECF system (Mauder et al., 2006). Although the YXASFT possesses a lack of field precipitation observations, we can speculate that precipitation may have occurred on 2016/02/07 based on a 1.8 ℃ drop in the air temperature and an increase of 13% in the relative humidity within the daily mean."

We accepted your suggestions and compared the mean variables (30 min averaged wind, temperature and humidity) obtained from the fast response instruments and

slow response instruments. As we can see from Fig.1, the two sensors could accurately measure the temperature and wind speed, and both were not affected by the precipitation. But, in term of water vapor observation, the fast response sensor EC150 made by CSI was obviously disturbed by the precipitation, and the data will be seriously polluted. In the period of comparison, four times of possible precipitation was marked by dashed ellipse in Fig.1, the time of four possible precipitation were on 2016/02/03, 2016/02/07, 2016/02/25, 2016/03/26, respectively. Strictly speaking, the ECF data in these four time windows must be eliminated before compared with COARE3.0. In this paper, we didn't eliminate the possible data polluted by precipitation, but it almost does not affect the validation. The LHF comparison between eddy covariance and COARE3.0 (Fig.3) shows a good consistency except for the above mentioned possible precipitation windows. We agree very much that if the ECF data during precipitation days were eliminated, the comparison between CAORE3.0 and ECF will be more consistent, which will further demonstrate the reliability of the COARE3.0 algorithm in SCS. In the next study, we will install the precipitation observation equipment on the YXASFT to improve the reliability of ECF data.

[Figure]

Fig.1 Time series of observed wind speed (ws), air temperature (ta), air relative humidity (rh) by the slow and fast response sensors, respectively. The time windows for possible precipitation were marked by dashed ellipse.

[revised manuscript text omitted]

(宋体)，（中文）中文(中国)

(宋体)，（中文）中文(中国)

[Figure]

Figure 43

[Figure]

**Figure 54**

[Figure]

**Figure**

[Figure]

[Figure]

[Figure]

**Figure 67**

[Figure]

**Figure 78**

(宋体),（中文）中文(中国)

(宋体),（中文）中文(中国)

[Figure]

Figure 89

[Figure]

[Figure]

**Figure** 10

（宋体），（中文）中文(中国)

（宋体），（中文）中文(中国)

[Figure]

[Figure]

**Figure 11**

---

## Referee Report (RR1)

From the author's response, they have answered all the questions raised by my first round of review, received my comments and made corresponding modifications in the revised manuscript. I have no more specific comments.

In the paper "Evaluation of OAFlux datasets based on in situ air-sea flux tower observations over the Yongxing Islands in 2016" the authors introduce the system design of a high quality air-sea flux tower (YXASFT) in the SCS, and also describe the air-sea interface observations in the Yongxing Island (SCS) for a long time period (a whole year). This area is characterized by a deep depth (around 1000m), and such data are very important for understanding the air-sea energy exchange in deep sea area. Given the importance of the OAFlux reanalysis datasets in the air-sea interaction research community, they evaluated the OAFlux datasets comprehensively using observations and provided some new conclusions. For example, they found that the reliability of OAFlux reanalysis data varies seasonally in the SCS, and gave suggestions on how to select OAFlux parameters in different seasons. Moreover, the authors analyzed the possible factors affecting the biases of OAFlux heat flux in different seasons.

Overall, this paper is well written with clear ideas and conclusions, the data observed from YXASFT are very rare and variable for air-sea interaction research in the SCS, and the content of this paper focuses on observation technology and data analysis, which is suitable for publication in the prestigious AMT.

So my conclusion is: this paper should be accepted and published in the final publication in AMT.

Other points:
P.24, P.25 In Figure 5,6, you have four panels on one figure with the same time scale in the X axis, I suggest to merge the four X axis into one, that will make the figure clearer.